# Distinct oligomeric assemblies of STING induced by non-nucleotide agonists

Anant Gharpure[1,3], Ariana Sulpizio[2,3], Johannes R. Loeffler[1], Monica L. Fernández-Quintero[1], Andy S. Tran [1], Luke L. Lairson [2] ✉ & Andrew B. Ward [1] ✉

STING plays essential roles coordinating innate immune responses to processes that range from pathogenic infection to genomic instability. Its adaptor function is activated by cyclic dinucleotide (CDN) secondary messengers originating from self (2'3'-cGAMP) or bacterial sources (3'3'-CDNs). Different classes of CDNs possess distinct binding modes, stabilizing STING's ligand-binding domain (LBD) in either a closed or open conformation. The closed conformation, induced by the endogenous ligand 2'3'-cGAMP, has been extensively studied using cryo-EM. However, significant questions remain regarding the structural basis of STING activation by open conformation-inducing ligands. Using cryo-EM, we investigate potential differences in conformational changes and oligomeric assemblies of STING for closed and open conformation-inducing synthetic agonists. While we observe a characteristic 180° rotation for both classes, the open-LBD inducing agonist diABZI-3 uniquely induces a quaternary structure reminiscent but distinct from the reported autoinhibited state of apo-STING. Additionally, we observe slower rates of activation for this ligand class in functional assays, which collectively suggests the existence of a potential additional regulatory mechanism for open conformation-inducing ligands that involves head-to-head interactions and restriction of curved oligomer formation. These observations have potential implications in the selection of an optimal class of STING agonist in the context of a defined therapeutic application.

The cyclic GMP-AMP synthase (cGAS)-stimulator of interferon genes (STING) pathway plays essential roles in innate immune responses, based on its ability to mediate cellular defense responses to pathogenic sources of both microbial and host-derived DNA. Aberrant intracellular DNA, resulting from infection or cellular damage, activates the enzyme cGAS, which catalyzes the synthesis of the second messenger cyclic dinucleotide (CDN) 2'3'-cyclic-di-GMP-AMP (cGAMP) from ATP and GTP[1–5]. STING is an ER membrane resident adaptor protein that binds cGAMP, as well as bacterially derived CDNs that include 3'3'-cyclic-di-GMP (cdG)[6,7] and 3'3'-cyclic-di-AMP (cdA)[7], which

triggers its phosphorylation, oligomerization, and translocation to the Golgi[8,9]. Although the functional necessity of STING's translocation to the Golgi has been called into question[10], CDN-induced STING oligomerization is essential for the recruitment of TANK-binding kinase 1 (TBK1) and downstream activation of interferon regulatory factor 3 (IRF3)[11]. STING-dependent phosphorylation and activation of these proteins ultimately culminates in the induction of IRF3- and NFκB-dependent transcription of type I interferons, such as IFNβ, and proinflammatory cytokines, such as TNFα, respectively[12,13]. As a central node in this pathway, STING has garnered significant interest as a drug

[1]Department of Integrative Structural and Computational Biology, Scripps Research, La Jolla, CA, USA. [2]Department of Chemistry, Scripps Research, La Jolla, CA, USA. [3]These authors contributed equally: Anant Gharpure, Ariana Sulpizio. ✉e-mail: llairson@scripps.edu; andrew@scripps.edu

target for a wide variety of pathologies, including infectious disease, autoimmune disorders, and cancer[14–17].

Importantly, the cGAS-STING pathway has been demonstrated to play an essential role in the promotion of anti-tumor immunity[18,19]. Intratumoral injection of cGAMP[20,21], or systemic administration of metabolically stable synthetic STING agonists[22–24], has been shown to inhibit tumor growth in a variety of syngeneic mouse tumor models by promoting STING/IRF3 signaling[18–20,25,26]. STING activation can also induce antitumor effects that are reliant on the NF-κB pathway. For example, STING activation in tumor-associated myeloid cells results in TNFα-mediated tumor endothelial cell apoptosis and the disruption of tumor vasculature[27]. Despite the clear potential of STING as an anti-tumor immunity promoting therapeutic target, translation of STING agonists into the clinic has been met with challenges, which are likely associated with narrow therapeutic indices and the potential promotion of NF-κB-related tumorigenic mechanisms[28,29]. A potentially relevant hypothesis in this context is that the relative strength of activation of the IRF3 versus NF-κB downstream signaling arms may impact anti-tumor activity versus tumorigenic and tolerability issues[30,31]. While an understanding of molecular mechanisms that might bias the relative levels of activation of these downstream pathways is currently lacking, one hypothesis is that these processes are differentially impacted by the class of agonist (i.e., cGAS- versus bacterially-derived) and the associated differences in their mode of binding to STING.

Structural studies of STING have revealed that it can adopt two distinct conformational states in response to different agonist classes. Crystal structures of the isolated ligand-binding domain (LBD) of STING reveal that this domain forms a butterfly-shaped dimer in an open position[32–35]. cGAMP binding induces substantial conformational change in the LBD, as the tips of the wings contract and a lid comprised of a four-stranded antiparallel beta sheet becomes ordered above the ligand (closed position)[3,36]. Interestingly, the related CDN STING agonist, cdG, does not promote the closed position and the LBD remains open upon agonist binding[32–35], potentially suggesting a distinct mechanism of activation. Recent progress in the development of stable non-nucleotide STING agonists has led to the identification of compounds such as SR-717[23], MSA-2[24], and linked amidobenzimidazoles (diABZIs)[22] that can be administered systemically. Co-crystal structures of these compounds with the STING LBD reveal that SR-717 and MSA-2 induce a cGAMP-like conformational change in the isolated LBD[23,24], whereas diABZI compounds retain the open conformation[22]. The structural mechanisms underlying full-length STING activation by cGAMP have been well-characterized by cryo-EM, however it remains unclear how non-nucleotide agonists, and specifically open LBD conformation-inducing ligands, drive oligomerization and activation.

Critical to the function of STING is its oligomeric state. Full-length STING exists as an obligate dimer with an N-terminal transmembrane domain (TMD) and a C-terminal LBD separated by a connector helix[37]. cGAMP binding prompts a drastic 180° rotation of the LBD relative to the TMD[37,38]. This rearrangement allows STING dimers to pack in a side-by-side fashion, facilitating the formation of high-order oligomeric chains. This quaternary structural arrangement is believed to be necessary to recruit TBK1 and stimulate downstream signaling pathways[11]. Notably, two additional small molecule ligands, C53 and NVS-STG2, have been identified, which each bind novel sites in the transmembrane domain of STING to promote the formation of these higher order oligomers[38–40]. It was also recently reported that inactive apo chicken STING, with an open LBD, forms double-stranded oligomers mediated by head-to-head and side-by-side interactions in the LBD[41]. This distinct oligomeric state was proposed to represent an autoinhibited conformation that potentially blocks the binding of TBK1 and translocation of STING to the Golgi. Additionally, there was a recent structure in complex with HB3089, a modified diABZI ligand[42];

however, in this study they only reported the dimeric form of STING, leaving open questions about whether STING forms high-order oligomers in the presence of an open-LBD ligand.

Here, we explore the conformational and oligomeric states of full-length human STING in the presence of closed-LBD (SR-717) and open-LBD (diABZI-3) non-nucleotide agonists, lending insight into potentially distinct modes of activation. Our structure of SR-717-bound closely resembles that of cGAMP-bound STING, which undergoes a 180° conformational change in the LBD that facilitates side-by-side packing into higher order STING oligomers. Alternatively, diABZI-3-bound STING forms distinct oligomeric assemblies that contain both side-by-side and head-to-head packing. This structural heterogeneity combined with observed kinetic differences in our cell-based assays lead us to speculate that additional regulatory mechanisms may be involved in open-LBD agonist-dependent STING activation.

## Results
### SR-717 complex

We expressed and purified full-length human STING and incubated the protein with SR-717 and C53[39], a TMD-binding allosteric ligand that has been that has been demonstrated to promote STING oligomerization[16,38,40]. Importantly, reported structures of STING in complex with cGAMP in the presence or absence of C53 show no significant structural differences, which indicates that inclusion of C53 does not alter the structure of STING-orthosteric ligand complexes[38,40,41]. Single-particle cryo-EM analysis of this sample revealed that STING molecules had assembled into curved oligomers (Supplementary Fig. 1A), akin to what was seen in previous structures complexed with cGAMP[41] and cGAMP/C53[38]. Local refinement using a mask around two adjacent STING dimers yielded a final reconstruction with a global resolution of 3.1 Å (Fig. 1A, Supplementary Fig. 1B, C). This map showed clear density for all ordered regions of the TMD and LBD and allowed for modeling of both ligands (Supplementary Fig. 2A, B). Two copies of SR-717 are found in each orthosteric site in the LBD, with the ligands and surrounding protein displaying near two-fold symmetry (Fig. 1B, C). As expected, C53 binds at the luminal side of the TMD, with asymmetric density that allows for fitting of the C-shaped ligand in the same orientation as seen in the cGAMP/C53 structure[38] (Supplementary Fig. 2B, F).

Our cryo-EM structure corroborates the co-crystal structure of SR-717 with the isolated STING LBD[23]. The two models overlay with an RMSD of ~1 Å, with the main divergence lying in the α2–α3 loop, which undergoes a conformational change in activated full-length oligomers to facilitate side-by-side packing (Supplementary Fig. 2C). The orientations of both SR-717 molecules are identical to those in the crystal structure (Fig. 1B, C, Supplementary Fig. 2C). The SR-717 pair is stabilized in the binding pocket by the sidechains of Y167 and R238 from either monomer, which form π-stacking and cation-π interactions with the pyridazine rings of the ligands respectively (Fig. 1C). Furthermore, the carboxyl groups of the ligands interact with the side chains of H232, R238, and T263 via polar interactions (Fig. 1B).

Overall, STING bound to SR-717 adopts the canonical closed position. The tips of the α1 helices in the LBD are contracted by ~16 Å when compared to full-length apo human STING[37] (Supplementary Fig. 2D). The lid, comprised of two beta strands from either monomer, is fully ordered above the ligand binding pocket, allowing H232 and R238 to interact with SR-717 (Fig. 1B, C). The α2–α3 loop in the LBD undergoes an inward displacement necessary to allow for backbone interactions from Q273 and S275 with the neighboring STING dimer to mediate the formation of high-order oligomers (Supplementary Fig. 2D, E). The LBD undergoes a 180° rotation relative to the TMD, which is believed to be a hallmark of STING activation[37,38]. The TMD is expanded when compared to apo STING and is consistent with the TMDs of STING in complex with cGAMP[41] and cGAMP/C53[38]. Here, the neighboring TMDs of the two molecules create an extensive

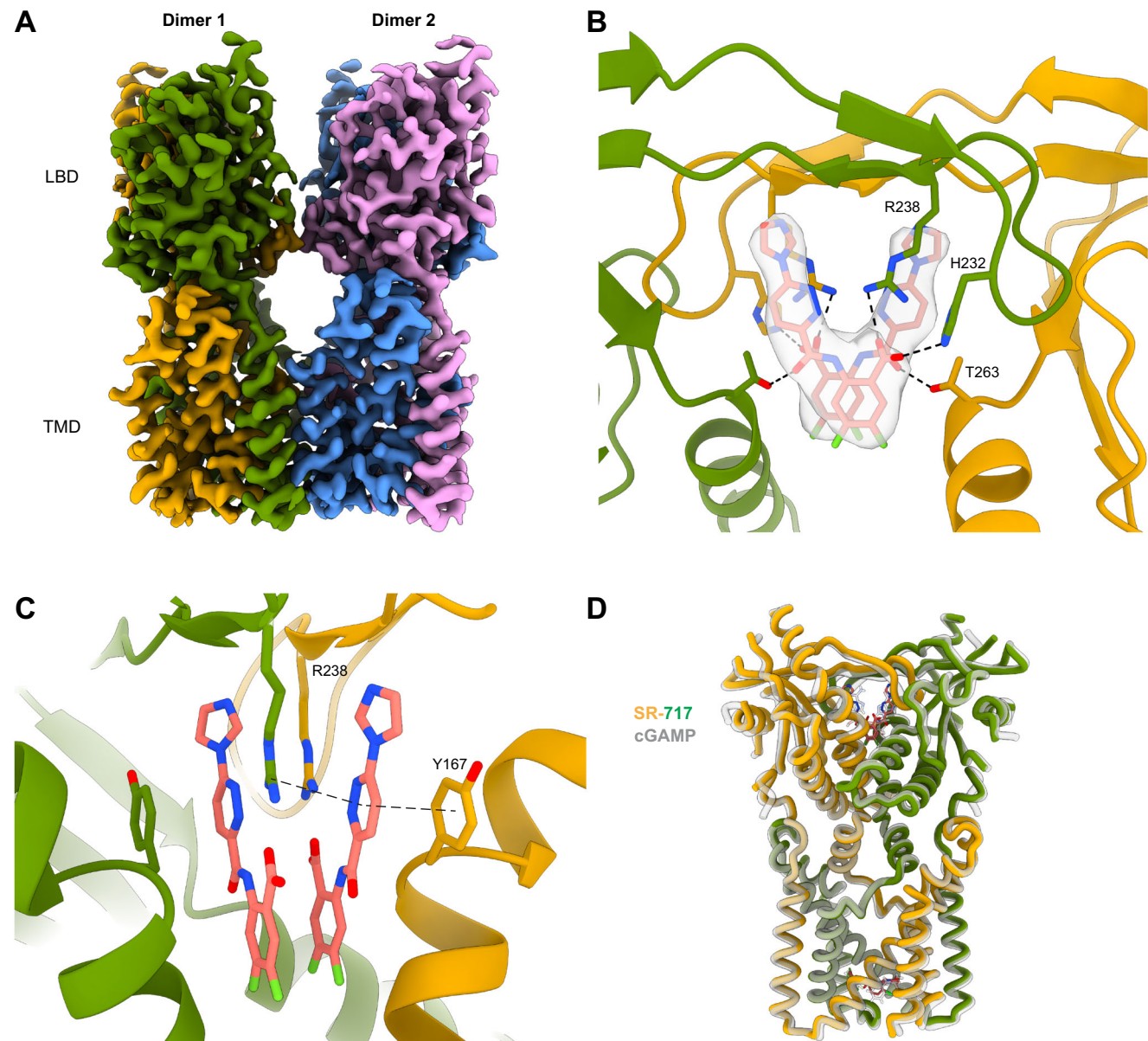

**Fig. 1 | SR-717 complex. A** Cryo-EM map of STING bound to SR-717. **B** SR-717 with associated density and interacting residues shown as sticks. **C** Side view of SR-717 binding pocket with interacting residues. One set of π-π and cation-π interactions shown in dotted lines. **D** Superposition of SR-717 complex (colored) and cGAMP complex (transparent gray; PDB ID: 7SII).

hydrophobic interface, with TM3 from either STING dimer interacting with TMs 1, 2, and 4 from the neighboring molecule (Supplementary Fig. 2F). The majority of the buried surface area within this interface is at the luminal side of the TMD, which helps promote the curvature of the oligomers[38]. Globally, this structure is consistent with the STING + cGAMP/C53 cryo-EM structure, with an RMSD of <1 Å, suggesting a conserved structural mechanism of activation by ligands that induce the closed position in the LBD (Fig. 1D).

**diABZI-3 complex**

To gain structural insight into the mechanism of STING activation by open conformation-inducing LBD agonists, we employed a similar strategy as described above using a diABZI compound instead of SR-717. We used diABZI compound 3 (diABZI-3), again with the inclusion of C53. diABZI-3 is modified from the original amidobenzimidazole STING ligand through a four-carbon linker between two copies of the molecule and a morpholinopropoxy group attached to one copy to

create an asymmetric dimeric ligand[22]. Surprisingly, in contrast to what was observed for SR-717 complexes, we found that particles in the cryo-EM micrographs formed long chains of double-stranded oligomers without any systematic curvature (Fig. 2A). Initial 3D processing of this dataset revealed two predominant classes of STING molecules (Supplementary Fig. 3A). One class, which we will refer to as the curved conformation, showed four copies of STING oligomerizing laterally with inherent curvature, similar to the SR-717 structure described above. However, there was also diffuse density for a second layer of STING molecules packing in a head-to-head manner with the curved oligomer, confirming the double-stranded oligomers seen in 2D classes (Fig. 2A, C).

In the second class, we found a bilayer of STING molecules devoid of curvature. The LBDs from both layers were well-resolved, but there was heterogeneity in the TMDs, which lacked clear density. Using 3D classification, we parsed out two distinct conformations from this class (Supplementary Fig. 3A). In the first conformation, two copies of

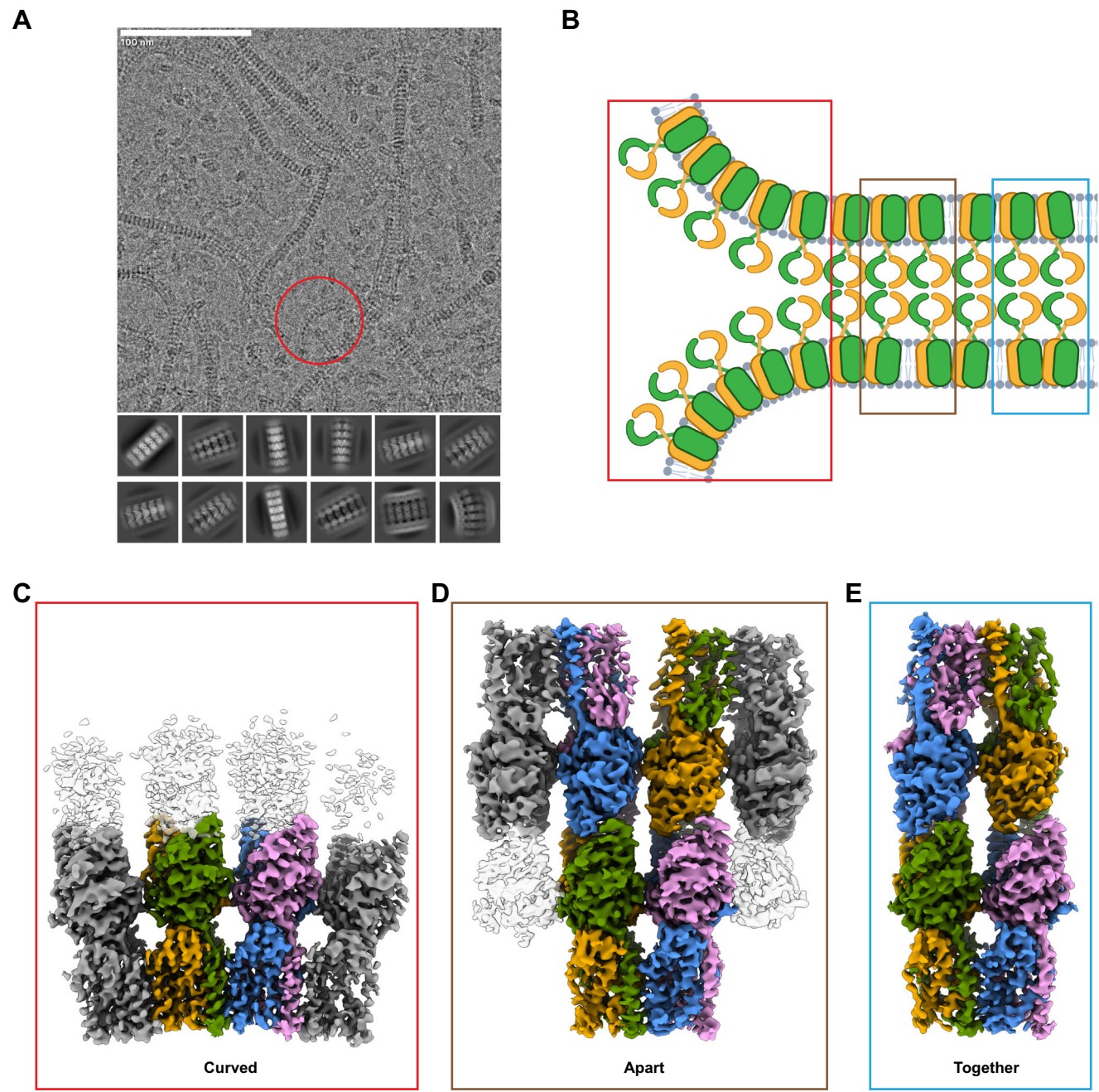

**Fig. 2 | diABZI-3 complex. A** Representative micrograph and 2D classes from STING-diABZI-3 dataset. Representative area of high curvature highlighted in red circle. **B** Cartoon model of diABZI-3-bound STING oligomer (Created in BioRender. Gharpure, A. (2025) https://BioRender.com/f06t201). **C** Curved conformation of STING-diABZI-3 complex. **D** Apart conformation of STING-diABZI-3 complex. **E** Together conformation of STING-diABZI-3 complex.

STING are present on either side (Fig. 2E). In this structure, which we will refer to as the together conformation, the neighboring dimers on both sides of the bilayer form interactions in the LBD and TMD. The lateral oligomeric interface is mediated by the same structural motifs as the SR-717 structure, namely through backbone hydrogen interactions from Q273 and S275 in the LBD α2–α3 loop and hydrophobic interactions in the TMD (Supplementary Fig. 4G, H). Importantly, the residues that had been identified in the cGAMP/C53 structure[38] as forming key interactions at the TMD interface, including L26, L30, L44, V48, L101, Y104, F105, and L109, also facilitate TM interactions in the diABZI-3 and SR-717 structures. We also observe density for C53 at the base of the TMD and modeled the ligand in the same orientation as the higher-resolution SR-717 structure (Supplementary Fig. S4B, H). In the second conformation (the apart conformation), there are six fully

resolved STING molecules (Fig. 2D). On one side, two STING molecules form a dimeric unit with an interface like that of the together conformation. On the opposing side, the central STING molecules still interact via the α2–α3 loop in the LBD, but the TMDs are splayed apart and instead interact with their other lateral neighbors.

Taken together, these structures illustrate the overall dynamic arrangement of this complex (Fig. 2B). STING, when bound to diABZI-3 and C53, oligomerizes laterally through LBD and TMD interactions similar to those seen in the SR-717/C53 and cGAMP complexes (Supplementary Fig. 4G, H). The protein also forms an extensive head-to-head interface with an equivalent and opposing chain. These head-to-head interactions partially restrict the ability of STING to form long oligomers with a high degree of curvature and instead promote a flatter double-layered quaternary structure. To maintain this linear

arrangement, dimeric units of STING are intermittently separated by breaks in TMD interactions as seen in the apart structure (Fig. 2B, D). 3D Variability Analysis (3DVA)[43] using particles from the together and apart classes supports this observation, showing continuous movement between those states with STING copies rocking back and forth to alternate TMD interaction partners (Supplementary Movie 1). Sporadically, areas of high curvature emerge, effectuated by particles from the curved conformation (Fig. 2A, B). This rounded shape is incompatible with a rigid bilayer, so the positions of STING molecules from the opposing layer may be more heterogeneous or display partial occupancy, leading to the observed weak density in the curved conformation.

To further investigate the dynamic heterogeneity of the STING-diABZI-3/C53 complex, we used 3DVA on particles from the curved conformation. The resulting volume series revealed a continuous spectrum of curvature (Supplementary Movie 2). The flatter particles showed density for an opposing layer of STING, which was largely absent in the more curved particles. Using the clustering method in 3DVA Display, we identified two distinct particle sets from the curved conformation representing these two endpoints- a cluster organized in a relatively flatter oligomer with density for the bilayer (cluster 1), and a monolayer cluster with pronounced curvature (cluster 2) (Fig. 3A). These results suggest that in the STING oligomer, the double-stranded assembly begins to weaken in areas of high curvature and is eventually disrupted entirely, yielding a curved monolayer of agonist-bound STING molecules (Fig. 2B).

It has been proposed that the positive curvature induced by STING agonists may play a role in downstream activity by enriching activated STING oligomers at the ER membrane ridges and helping to engage COPII machinery for transport to the Golgi[38,41]. It is therefore possible that differences in curvature of the oligomers may lead to differences in function. As our diABZI-3 dataset produced a range of conformational states, we sought to quantify and compare the degree of curvature of these states to that induced by a closed-LBD agonist such as SR-717. To do so, we used maps that contained four adjacent STING molecules and measured the angle between the planes that define the terminal copies. The SR-717 complex induced an angle of 26.4° (Fig. 3B). Similarly, cluster 2 from the curved conformation of diABZI-3, which was resolved as a monolayer, produced an angle of 23.4° (Fig. 3C). The related cluster 1 was slightly less rounded at 21.2° (Fig. 3D), and the four adjacent copies from the apart conformation displayed nearly no curvature, with a measured angle of 2.6° (Fig. 3E). These results suggest that while diABZI-3 can induce similar curvature to closed-LBD agonists such as SR-717 (Fig. 3C), the constraints from the bilayer may partially restrict the ability of diABZI-3-bound STING to form curved oligomers. The flattened diABZI-3-bound STING molecules in the bilayer (Fig. 3E) may prevent COPII recruitment and anterograde transport which could play an additional regulatory role in a mechanism that serves to modulate rates of STING activation and/ or the strength of downstream pathway signaling by this class of ligands.

We evaluated ligand class-dependent differences in relative rates of STING activation, by assessing STING (S366) phosphorylation status in THP-1 cells (Fig. 3G, H). When evaluated at -EC$_{80}$ concentrations that induce the same degree of pathway activation, diABZI-3 was found to activate STING at a slower rate, with peak levels of STING phosphorylation occurring -1 hour later for diABZI-3 when compared to SR-717 (2–4 hours versus 1–2 hours of stimulation, Fig. 3G, H). Given the disparity in cell-based potencies and limitations of solubility at superphysiological concentrations of the dimeric diABZI-3 ligand, the evaluated concentrations in this assay are physiologically most relevant. Nonetheless, the potential caveat of these concentration differences should be noted. However, a consistent relative delay in downstream pathway activation was also observed, based on induction of interferon regulatory factor (IRF)-dependent luciferase reporter signal.

When comparing 8 versus 24 hours of agonist stimulation, diABZI-3 is -100-fold less potent (EC$_{50}$ = 14 nM versus EC$_{50}$ = 0.11 nM, Fig. 3I), whereas SR-717 is only -2-fold less potent (EC$_{50}$ = 1.3 μM versus EC$_{50}$ = 0.7 μM, Fig. 3I). Importantly, this delayed activation is also observed in these functional assays when C53 is included (Supplementary Fig. 5). These observed differences could be consistent with an additional regulatory mechanism for open conformation-inducing ligands involving the observed head-to-head interactions and restriction of curved oligomer formation.

## Head-to-head interface

It has previously been reported that STING can assemble in a bilayer mediated by head-to-head interactions in the LBD[41]. This oligomeric state was observed in apo chicken STING and was proposed to act as an autoinhibited conformation to prevent aberrant overactivation of the cGAS-STING pathway. The authors of that study suggested that this assembly could promote the retention of STING in the ER and prevent the association of TBK1 with its binding site at the top of the LBD. In this model, the 180° rotation of the LBD elicited by agonist binding would disrupt the head-to-head interface and release autoinhibition, allowing STING to form curved oligomers and translocate to the Golgi.

We were therefore surprised to find that diABZI-3-bound STING particles also predominantly assembled into a bilayer mediated by head-to-head interactions. However, the diABZI-3 complex is structurally distinct from the reported autoinhibited bilayer. In the autoinhibited structure, the lid is ordered in an asymmetric manner with one chain resembling the lids seen in SR-717 and cGAMP structures and the other in a non-canonical fold with the two strands flipped (Fig. 4A). The head-to-head interface is mediated by a conserved LP motif (L225 and P226 in human STING) that docks into a hydrophobic pocket comprised of the LBD α1 helix and the base of the lid region on the opposing layer (Supplementary Fig. 6A, B). In contrast, the diABZI-3 structure has an unresolved lid and the LP motif is not situated closely to the opposing layer (Fig. 4C, Supplementary Fig. 6C-D). Instead, the head-to-head interaction appears to be facilitated by the loop following the LBD α1 helix which interacts with the β2 and β3 strands as well as the post-β3 loop of the opposite layer (Fig. 4E–G). Notably, we resolved additional residues following the α1 helix that were unresolved in the SR-717 and cGAMP complexes including N187 and N188. This post-α1 loop, which also includes H185 and Y186, extends laterally along the head-to-head interface, potentially creating a polar scaffold for the interface with the opposing STING molecule.

Comparison of the together and apart conformations reveal that the head-to-head interfaces between different copies of STING are quite heterogeneous and asymmetric (Supplementary Fig. 3C, D, Supplementary Fig. 6E). 3DVA confirms this finding, showing that this region acts as a hinge that loosely anchors the bilayer assembly, but allows individual STING molecules to swing along the lateral axis (Supplementary Movie 1). Consequently, the density at this interface did not allow for confident modeling of interacting sidechains. We therefore employed molecular dynamics (MD) to study the details of the head-to-head interface. MD simulations from the together and apart structures confirmed that the head-to-head interactions were predominantly mediated by residues located on the post-α1 loop, β2 and β3 strands, and the post-β3 loop. As predicted, the four residues from the post-α1 loop were involved in a network of contacts with the opposing STING molecule, interacting with residues near the lid such as R220, F221, D223, as well as residues in the post-β3 loop including E246, Q252, and R253 (Fig. 4G).

Further comparison of the autoinhibited head-to-head interface with that of the diABZI-3 structures reveals that it buries nearly twice as much solvent-accessible surface area with an average of 1258 Å$^2$ compared to 664 Å$^2$ for the together state and 589 Å$^2$ for the apart state (Fig. 4B, D). This may indicate a relatively weakened trans-interface in diABZI-3-bound STING, consistent with the observation of

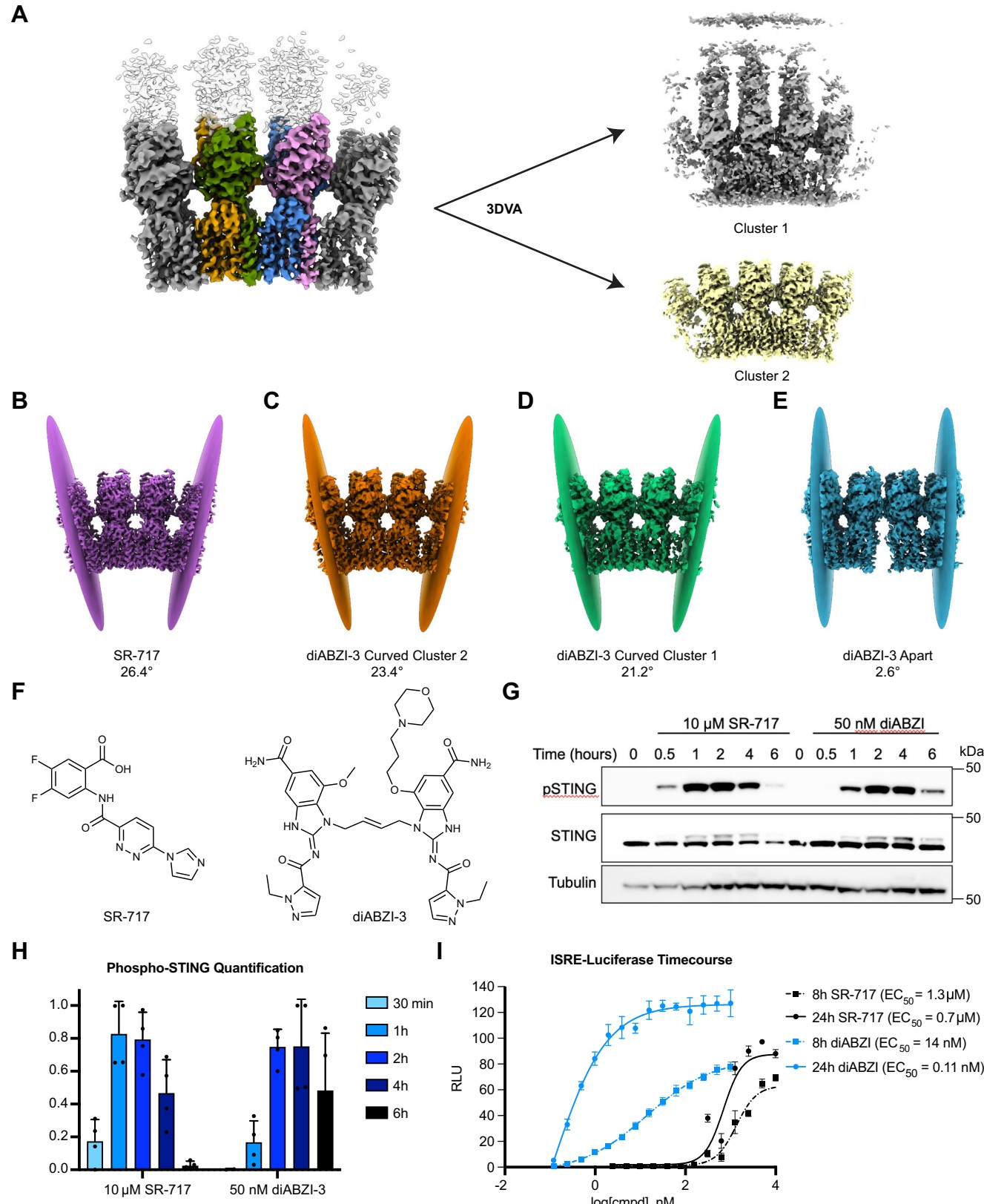

**Fig. 3 | diABZI-3 heterogeneity and delayed activation. A** 3DVA clustering of curved conformation. **B** Angle of curvature of SR-717 complex. **C** Angle of curvature of diABZI-3 curved cluster 2. **D** Angle of curvature of diABZI-3 curved cluster 1. **E** Angle of curvature of diABZI-3 apart. **F** Chemical structures of SR-717 and diABZI-3. **G** Time course of STING phosphorylation following stimulation with SR-717 and diABZI-3 at $EC_{80}$ concentrations. **H** Quantification of phospho-STING western blots normalized to the loading control. **I** ISRE-Luciferase dose response for SR-717 and diABZI-3 at 8-and 24-hours post-stimulation. In (**H**) data are shown as mean ± SD from $n = 4$ independent experiments. In (**I**) data are shown as mean ± SD from $n = 3$ technical replicates. Data presented in (**G**) and (**I**) are representative of $n = 4$ and $n = 3$ independent experiments, respectively. Source data are provided as a Source Data file.

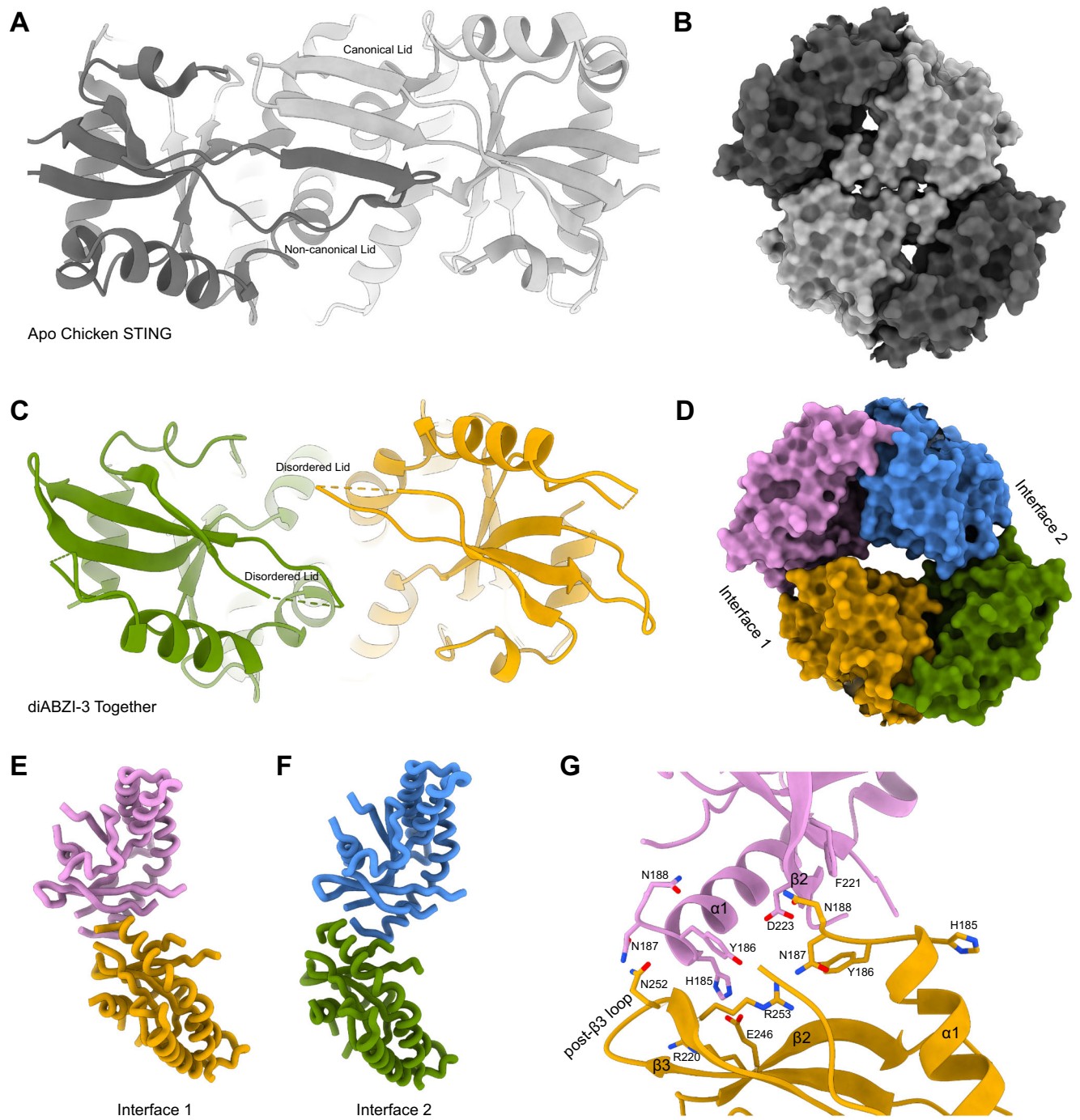

**Fig. 4 | Head-to-head interactions. A** Top view of Apo Chicken STING (PDB ID: 8IK0) showing ordered lid. **B** Surface representation of head-to-head interacting LBDs in Apo Chicken STING (PDB ID: 8IK0). **C** Top view of diABZI-3 together conformation. **D** Surface representation of head-to-head interacting LBDs in diABZI-3 together conformation. **E** Cartoon representation of interface 1 of the diABZI-3 complex from panel **D**. **F** Cartoon representation of interface 2 of the diABZI-3 complex from panel **D**. **G** Residues identified by molecular dynamics as contributing to the head-to-head interaction mapped onto the experimental structure at interface 1.

monolayered oligomers in the sample. It is important to note that these two sets of structures came from different STING orthologs and a structure of apo human STING in an autoinhibited conformation would provide a better reference for comparison.

**diABZI-3 LBD and connector**
Within the LBD we observed clear density for the dimeric amido-benzimidazole moiety corresponding to the core of diABZI-3 ligand along with asymmetric density for the morpholinopropoxyl tail

(Fig. 5A, Supplementary Fig. 4A, Supplementary Fig. 7). This suggests a preferential binding orientation of the asymmetric ligand in the LBD, which is an interesting observation considering that STING is a homodimeric protein that should be able to bind asymmetric molecules in two equivalent orientations[3]. However, the functional implications of ligand directionality remain unclear, and it is possible that there is a mixture of both orientations in each LBD.

The diABZI-3 compound is stabilized in the binding pocket by a series of aromatic residues, namely Y163, Y167, and Y240, and forms

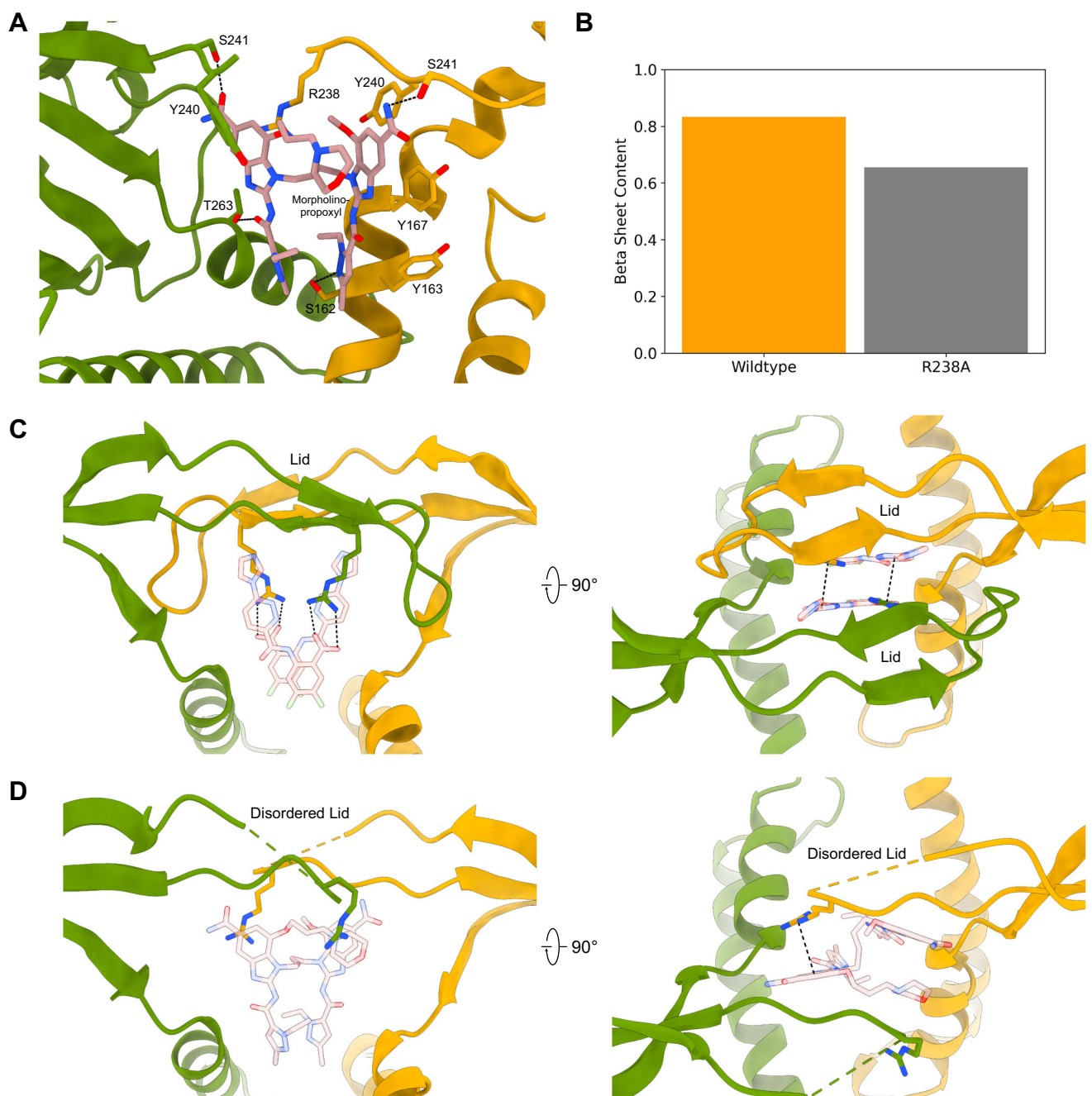

**Fig. 5 | diABZI-3 LBD. A** diABZI-3 with associated density and interacting residues shown as sticks. Hydrogen bonds are indicated with dashed lines. **B** Graph showing beta sheet content in MD simulations of WT STING and R238A mutant in SR-717 complex. **C** LBD of SR-717 complex with ordered lid. Ligand and R238 are shown as sticks. Side view shows electrostatic interactions and top view shows cation-π interactions. **D** LBD of diABZI-3 complex with disordered lid. Ligand and R238 are shown as sticks. Side view shows electrostatic interactions and top view shows cation-π interactions.

hydrogen bonds with S162, S241, and T263 (Fig. 5A). As previously reported in the crystal structure with diABZI compound 2 (a symmetric linked amidobenzimidazole without the morpholinopropoxyl group)[22] and the HB3089 cryo-EM reconstruction[42], the lid is largely unresolved with a lack of clear density between residues Q227 and R238 (Fig. 5D).

Notably, sidechain density for R238 is present in only one chain of each dimer, where the guanidinium group appears to be poised to form a cation-π interaction with a benzimidazole ring of diABZI-3. The sidechain of R238 from the other chain cannot occupy the same position, as it would sterically clash with the morpholinopropoxyl group of diABZI-3 (Fig. 5D). Interestingly, density for R238 is only observed in a subset of STING-diABZI-3 complexes between the

curved, together, and apart structures. This is in contrast with structures of STING in complex with closed conformation-inducing ligands such as SR-717 and cGAMP[3], where R238 residues provide important symmetrical interactions that are critical for ligand binding (Fig. 5C). Additionally, the mouse STING-specific ligand, DMXAA also induces the closed conformation and utilizes key hydrogen bonds from both R237 sidechains (the murine equivalent to R238) to stabilize the ligand in the binding pocket[36]. Conversely, R238 does not show essential interactions with other open conformation inducing ligands such as cdG[32] and HB3089[42]. Thus, strong engagement of both copies of R238 with orthosteric ligands may be a determinant for inducing the closed conformation by helping to stabilize the natively (unliganded)

unstructured loop between Q227 and Y240 into the ordered anti-parallel beta sheet that comprises the lid. The constraints imposed by the formation of this beta sheet may also force the LBD into a more overall compact conformation seen in the closed conformation. To test this hypothesis, we used MD to compare the integrity of the lid of the SR-717 complex in WT STING and an R238A mutant. The R238A mutant reduced beta sheet content in the lid by 18%, suggesting that the interaction between ligands and the guanidinium group of this residue may indeed play a key role in maintaining an ordered lid and inducing a closed LBD (Fig. 5B). Additionally, the critical role of R238 in stabilizing SR-717 is further emphasized by a substantial increase in electrostatic interaction energy of 30 kcal/mol (R238-SR-717: -147 kcal/mol vs. R238A-SR-717: -117 kcal/mol).

The average distance between the α1 helices, as defined by the distance between the Cα atom of H185, in the together and apart conformations is 42.3 Å, which is roughly 10 Å shorter than in apo STING and 6 Å longer than the SR-717 complex (Supplementary Fig. 4C-D). This open conformation is also similar to HB3089 complex (44.5 Å; Supplementary Fig. 4E) but is contracted when compared to the crystal structure of the isolated LBD with diABZI compound 2 (52.7 Å; Supplementary Fig. 4F). The α1 helices in the curved conformation are slightly more contracted, with an average distance of ~40 Å. These observations are consistent with the idea that STING LBDs are slightly more constricted in the context of the full-length protein than they are as truncated LBDs[37,42], and also suggest that the head-to-head interactions may keep the LBD more open.

Our diABZI-3 structures also display the same 180° rotation of the LBD relative to the TMD that was seen in the SR-717 complex as well as in cryo-EM structures with cGAMP (Fig. 6A, B). This is in direct disagreement with what was reported for the HB3089 structure, where the connector helix and LBD α1 helix form a right-handed crossover (Fig. 6B). This crossover conformation, which has been attributed to represent an inactive state, was described by the authors of that study as a distinct mode of activation. However, it is possible that the lack of rotation between the LBD and TMD in the HB3089 complex is due to the lack of lateral oligomerization. In our structure, we see an inward displacement of the LBD α2–α3 loop to accommodate side-by-side packing of STING dimers, similar to what is seen in SR-717- and cGAMP-activated structures (Fig. 6C, D). The loop in this position would directly clash with the connector/LBD α1 helices in the crossover conformation (Fig. 6D), which could promote the LBD rearrangement. Thus, we speculate that the 180° rotation observed in agonist-bound structures is a direct result of the conformational changes associated with the activation and higher-order oligomerization of STING.

### Proton pore

It was recently reported that STING also functions as a proton channel[44,45]. STING activation can induce proton flux in the Golgi, initiating auxiliary functions of STING, including noncanonical light-chain 3B lipidation[46] and inflammasome activation[47]. The authors of these studies used a pore-prediction program to hypothesize that the proton channel lies along the dimeric interface in the TMD. The predicted pore is only present in the activated conformation of STING, which displays an expanded TMD when compared to the inactivated state. It was also shown that the C53 binding site overlaps with the proton channel and addition of this compound is sufficient to block proton flux.

Consistent with these findings, our MD simulations showed continuous water wires through the center of the TMDs in both SR-717 and diABZI-3 complexes (Supplementary Fig. 8A, C). Inclusion of C53 in these simulations disrupted the water wires, with C53 acting as a hydrophobic pore blocker (Supplementary Fig. 8B, D). Structurally, the pore can be divided into a hydrophilic section at the top of the TMD and an aromatic section towards the bottom (Supplementary Fig. 8E, F). The hydrophilic section inside the pore allows for the

formation of a long-lasting water hydrogen bond network between residues K137, N61, Q55 and Q128, facilitating the water wires and consequently supporting its function as proton channel. The lower part of the pore consists of mainly aromatic/hydrophobic residues, among which histidine (H50 and H42) and tyrosine (Y46 and Y106) residues are involved in water interactions. Y46 and H50 directly interact with C53 when the ligand is present, effectively blocking the water wire and sealing the pore (Supplementary Fig. 8F).

## Discussion

Potential differences related to the structural basis of activation by open versus closed STING LBD conformation-inducing ligands remains an outstanding question in the field. An understanding of how these differences might impact downstream mechanisms has potential implications in the context of targeting STING to promote anti-tumor immunity, adjuvant usage, and tolerability. Here, we use cryo-EM to gain insight into STING's oligomeric states and activation mechanisms in response to distinct ligand classes. Our structure of SR-717-bound STING reveals that the closed-LBD agonist induces STING oligomers that very closely resemble those formed upon binding of the endogenous ligand 2′3′-cGAMP[37]. These oligomers exhibit a characteristic curved conformation facilitated by lateral interactions between adjacent STING dimers.

In contrast, in the context of the open conformation-inducing LBD agonist diABZI-3, we observe the induced formation of a distinct oligomeric state mediated by both side-by-side and head-to-head contacts in the ligand bound state. Notably, the head-to-head oligomeric state observed for diABZI-3-bound STING is reminiscent of, but structurally distinct from, the previously characterized head-to-head oligomer identified with apo-STING[41]. Given that this state was classified as an autoinhibited form of STING, we propose a model in which the binding of an open conformation-inducing LBD ligand leads to an intermediate partially autoinhibited state, which is associated with an equilibrium between the inherent curvature induced by lateral oligomerization and linearity enforced by the head-to-head interactions. The constraints from the trans interactions produce a potentially inhibited double-layered assembly consisting of particles from the apart and together conformations. However, monolayered regions of the oligomeric chain with high curvature in the curved conformation of the diABZI-3 complex resemble the oligomers induced by closed-LBD agonists, suggesting a similar activation mechanism for a subset of particles.

We believe that the range of oligomeric states observed with diABZI-3 may be derived from a less effective ability of open-LBD ligands to fully disrupt the autoinhibited apo-oligomer. Furthermore, this head-to-head conformational arrangement of STING induced by diABZI-3 likely impedes the formation of curved oligomers, which could delay or modulate the tone or strength of downstream pathway activation. Consistent with this potential model, based directly on TBK1-dependent STING phosphorylation status and induced levels of IRF reporter signal, delayed rates of STING activation were observed for diABZI-3 when compared to SR-717 in functional assays. Importantly, observed kinetic differences could also directly result from the structural differences of the closed and open conformations of LBDs and indeed our observations could arise from a combination of both effects.

Previous structural studies have extensively characterized the closed binding mode of STING and determined that a 180° rotation in the LBD relative to the TMD, upon ligand binding, displaces the C-terminal tail and exposes patches that promote side-by-side packing, which facilitates an activated STING oligomerization state[37]. Cryo-EM studies of TBK1-STING indicate that oligomerization is essential for TBK1 autophosphorylation, as geometric constraints inhibit TBK1 from autophosphorylating in cis[11]. Instead, activation of TBK1 occurs through higher order STING oligomerization which allows for trans-

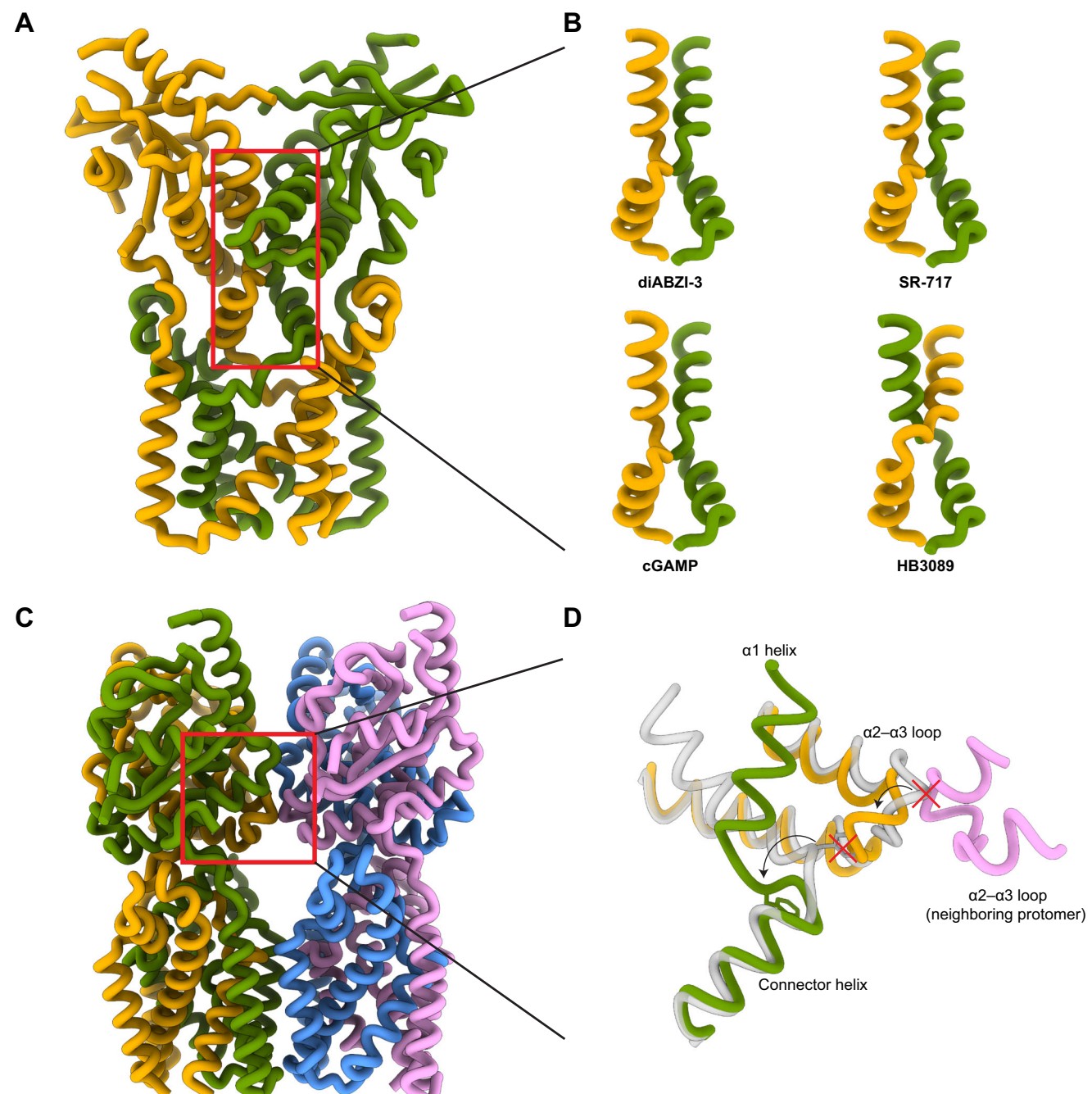

**Fig. 6 | Connector helix rotation. A** Model of the diABZI-3 dimer (together conformation). Box indicates region highlighted in panel (**B**). **B** Cartoon representation of connector and LBD α1 helices of STING-agonist complexes (PDB IDs: cGAMP-7SII, HB3089- 8GT6). **C** Model of diABZI-3 dimeric interface. Box indicates region highlighted in panel (**D**). **D** Closeup of dimeric interface with diABZI-3 complex shown in color and HB3089 (PDB ID: 8GT6) complex shown in gray. Red X's indicate clashes and arrows indicate movement necessary to avoid clashes.

autophosphorylation. Furthermore, activated TBK1 is unable to phosphorylate S366 of the same STING dimer, and instead must phosphorylate neighboring STING dimers. Therefore, given the well characterized role of STING oligomerization made possible by the closed-LBD conformational rearrangement, the way in which an open-LBD ligand induces STING activation remained an outstanding question in the field. Here, we find that open-LBD agonist diABZI-3 binding does in fact induce this 180° rotation. The formation of a closed lid within the LBD is not required for this 180° rotation that facilitates the formation of activated STING oligomers. In our structure, as is consistent with the cGAMP- and SR-717-bound structures, this rotation is crucial for side-by-side packing and higher order oligomerization.

Therefore, we believe that this conformational rearrangement, marked by a 180° rotation in the LBD, is required for the induction of pathway activation by both classes of ligands.

The observed structural heterogeneity of diABZI-3-bound STING oligomers combined with the delayed activation observed in our functional assays may indicate that differing stimuli can influence the strength and tone of downstream signal transduction. This is particularly interesting in the context of understanding potential differences associated with CDNs derived from self or pathogenic infection versus commensal bacteria, or when considering the translation of STING agonists into the clinic. Harnessing the therapeutic potential of STING will likely require mitigation of on-target toxicity, which could

require targeting approaches or specific activation of CD8 + T cell-mediated antitumor immunity by biasing or modulating downstream type I interferon signaling. In this light, further expansion of these analyses to a wide variety of STING agonists (i.e., synthetic agonists, CDNs, and non-LBD binders) will be required, in order to fully understand the functional relevance of the unique oligomeric state we observe here in the context of a synthetic open conformation-inducing LBD ligand. Ultimately, these findings highlight and contribute to the understanding of the diverse oligomeric landscape of STING and provide a blueprint for future studies examining the functional roles of different ligand classes.

## Methods

### Expression and Purification of hSTING
The coding sequence of hSTING 1-343 with a short C-terminal linker and Strep-tag (GGGSGGGSGGGSAWSHPQFEK) was codon-optimized and inserted into the pEZT-BM vector. HEK293F cells were transfected with 1 mg DNA and 3 mg PEI per liter of cells. Cells were harvested after 72 hours. Cells were resuspended in TBS (20 mM Tris 7.4, 150 mM NaCl) supplemented with 1 mM PMSF and 0.5 mM TCEP and lysed via sonication. Lysed cells were centrifuged at 3000 g for 10 min, and the resulting supernatant was centrifuged for 2 hours at 186,000 g to isolate the membrane fraction. Membrane pellets were stored at -80 °C until further use. Membrane pellets were homogenized and solubilized in TBS supplemented with 1% n-dodecyl-β-D-maltopyranoside (DDM), 0.2% cholesteryl hemisuccinate (CHS), 1 mM PMSF, 0.5 mM TCEP, and 1 mM $CaCl_2$ for 2 hours. Solubilized membranes were clarified by centrifugation for 40 min at 186,000 g, then passed over Strep-Tactin XT 4Flow (IBA) resin via gravity flow. The resin was washed with SEC buffer (TBS with 0.02% DDM, 0.004% CHS, and 0.5 mM TCEP), and protein was eluted with 10X Buffer BXT (IBA) diluted to 1X in SEC buffer. Eluted protein was concentrated and further purified using a Superdex 200 Increase 10/300 GL column (GE healthcare) in SEC buffer.

### Cryo-EM sample preparation and data collection
SEC-purified protein was pooled and diluted to 5 µM. For the diABZI-3 sample, protein was incubated with 30 µM diABZI compound 3 (Selleck) and 30 µM C53 (Cayman). For the SR-717 sample, protein was incubated with 100 µM SR-717 (Selleck) and 30 µM C53. Protein-ligand complexes were incubated overnight and then concentrated to 6-11 mg/mL. 3 µL of sample was applied to glow-discharged UltrAuFoil 1.2/1.3 300-mesh grids (Quantifoil). Grids were blotted for 3-4 s after a 3 s wait time and plunge frozen into liquid ethane using a Vitrobot Mark IV (ThermoFisher) operating at 4 °C and 100% humidity. Micrographs were collected on a Glacios 2 microscope (ThermoFisher) operating at 200 kV equipped with a Falcon 4i detector (ThermoFisher). Nominal magnification and pixel size were 190,000x and 0.718 Å respectively. Data were collected with EPU (ThermoFisher) with an approximate exposure dose of 45 e/Å$^2$ and a nominal defocus range of -0.6 to -1.5 µm. A total of 4818 micrographs were collected for the SR-717 dataset and 11729 micrographs were collected for the diABZI-3 dataset over two data collections.

### Cryo-EM data processing
Micrographs were aligned using Patch Motion Correction and CTF estimation was done by Patch CTF in CryoSPARC Live[48]. Micrographs with CTF fits worse than 8 Å were discarded. All following processing was done using CryoSPARC. Blob picker was used on an initial subset of several hundred micrographs for each dataset. Following 2D classification, classes resembling oligomeric STING were used to template pick all micrographs. False positives were removed via multiple rounds of 2D classification. For the SR-717 dataset, selected particles were further classified with Heterogeneous Refinement using PDB 7SII (low-pass filtered to 20 Å) as an initial model. A final subset of 272,478

particles was selected for Non-Uniform Refinement[49], Global and Local CTF Refinement, and Local Refinement with a mask around the central two dimers using C2 symmetry yielding the final map.

For the diABZI-3 dataset, particles selected after 2D classification were subjected to Heterogeneous Refinement using PDB 8IK0 (low-pass filtered to 20 Å) as an initial model. A class of 200,337 particles containing four adjacent STING copies with TMD interactions was selected for Global and Local CTF Refinement and Non-Uniform Refinement for the curved conformation. These particles were also subjected to 3D Variability Analysis[43] with clustering to identify bilayered and monolayered populations. An additional class from the Heterogenous Refinement that showed head-to-head interactions in the LBD was selected for further classification. 3D Classification was used to identify the together and apart conformations with 175,203 and 410,809 particles respectively. The final maps were resolved using Non-Uniform Refinement, Global and Local CTF Refinement, and Local Refinement.

### Model building and refinement
PDB 7SII was used as an initial model for all STING structures. The model was fit into experimental maps using UCSF ChimeraX[50] and morphed into the density using Phenix[51]. The models were manually adjusted using Coot[52,53] and further refined using real-space refinement in Phenix. Structural figures were made using UCSF ChimeraX and Pymol.

### Molecular dynamics
As starting models for our MD simulations, the cryo-EM structures of STING (presented in this study), as well as the recently published apo chicken STING oligomer (PDB 8IK0) were used. We modeled the missing residues in the cryo-EM structures using the modeling/building missing loops tool in Molecular Operating Environment (MOE)[54] and prepared the structures for the simulations using MOE[54]. For the head-to-head oligomers, we used one lipid bilayer, as we truncated the second molecule to the head to calculate the head-to-head interactions. Simulations were performed with and without C53 for the cryo-EM structures presented in this study. The C-terminal and N-terminal parts of each domain were capped with acetylamide and N-methylamide to avoid perturbations by free charged functional groups. The structure was aligned in the membrane using the PPM server[55] and inserted into a plasma membrane consisting of POPC (1-palmitoyl2-oleoyl-sn-glycero-3-phosphocholine) and cholesterol in a 3:1 ratio, using the CHARMM-GUI Membrane Builder[56,57]. Water molecules and 0.15 M KCl were included in the simulation box. All simulations of the WT and mutants were performed using GROMACS 2020.2[58,59] with the CHARMM36m force field for the protein, lipids and ions[60]. The TIP3P water model was used to model solvent molecules[61]. The system was minimized and equilibrated using the suggested equilibration input scripts from CHARMM-GUI[62], i.e., the system was equilibrated using the NPT ensemble for a total time of 2 ns with force constraints on the system components being gradually released over six equilibration steps. The systems were further equilibrated by performing a 10 ns simulation with no electric field applied. The temperature was maintained at T = 300 K using the Nosé-Hoover thermostat[63], and the pressure was maintained semi-isotropically at 1 bar using the Parrinello-Rahman barostat[64]. Periodic boundary conditions were used throughout the simulations. Long-range electrostatic interactions were modeled using the particle-mesh Ewald method[65] with a cut-off of 12 Å. The LINCS algorithm[66] was used to constrain bond lengths involving bonds with hydrogen atoms. Then molecular dynamics simulations were performed for 2 × 500 ns, with time steps of 2 fs, at 300 K and in anisotropic pressure scaling conditions. Van der Waals and short-range electrostatic interactions were cut off at 10 Å, whereas long-range electrostatics were calculated by the Particle Mesh Ewald (PME) method. The interaction energies were calculated using the energy function implemented in GROMACS. To

quantify the difference of water distributions along the z-axis of the pore with and without C53 present, we performed a Kernel-Density-Estimation (KDE) analysis of the water molecules in the pore[67].

## Tissue culture
ISG-THP-1 cells (InvivoGen, Cat# thpl-isg) were maintained in growth media consisting of RPMI 1640, 2mM L-glutamine, 25 mM HEPES, 10% heat-inactivated fetal bovine serum (FBS), 1000 units/ml penicillin, 1000 µg/ml streptomycin, and 0.25 µg/ml Amphotericin B.

## ISRE-luciferase assay
ISG-THP-1 cells were resuspended in low-serum growth media (2% FBS) at a density of $2 \times 10^5$ cells/mL. 50 uL of cells were plated per well of a 384-well white Thermo Fisher plate and incubated for the indicated times. For luciferase detection, 20 uL of Quanti-Luc (Invivogen) reagent was added to each well and luminescence was immediately read using an Envision plate reader. Luminescence readings were normalized to vehicle-treated samples, and therefore reported as relative light units (RLU).

## Western blotting
Cell pellets were resuspended in lysis buffer (25 mM HEPES, pH 7.4, 300 mM NaCl, 1.5 mM MgCl2, 1 mM EGTA, 1% NP-40, 1% sodium deoxycholate, 2.5 mM sodium pyrophosphate, 1 mM glycerophosphate) containing freshly added protease and phosphatase inhibitors (ThermoFisher Scientific). Cell lysates were clarified by centrifugation (10 minutes, 18,000 g, 4 °C) and the protein concentration was determined using the Pierce BCA kit. Normalized lysate was mixed with Bolt LDS sample buffer and Bolt reducing agent (ThermoFisher Scientific) and boiled for 10 minutes at 70 °C. Bolt 4-12% Bis-Tris gels and Bolt mini transfer system were used for western blotting (ThermoFisher Scientific). Gels were transferred onto a PVDF membrane and blocked in 5% milk in TBS-Tween (0.1%). Blots were incubated with primary antibody overnight at 4 °C in 5% BSA in TBS-Tween (0.1%) Antibodies used include: Tubulin (Sigma-Aldrich cat# T5326 1:5000, clone: GTU-88, Lot# 109M4784V), Vinculin (Thermo-Fisher cat# 14-9777-82 1:10,000, clone: 7F9, Lot# 2502412), phospho-STING (Cell Signaling Technology cat# 19781S 1:1000, Lot: 9), STING (Cell Signaling Technology cat# 13647 1:1000, Lot: 3). Membranes were incubated with anti-mouse or anti-rabbit-HRP in 5% milk in TBS-Tween (0.1%) for 1 hour at room temperature and then with Femto ECL (Thermo Fisher) for 1 minute before visualizing using a ChemiDoc Imager. Uncropped and unprocessed blots can be found in the Source Data file.

## Reporting summary
Further information on research design is available in the Nature Portfolio Reporting Summary linked to this article.

# Data availability
The atomic models and cryo-EM density maps generated in this study have been deposited to the PDB and EMDB respectively. The accession numbers are 9CT3 and EMD-45897 (SR-717), 9CT4 and EMD-45898 (diABZI-3 Curved), 9CT6 and EMD-45900 (diABZI-3 Apart), and 9CT5 and EMD-45899 (diABZI-3 Together). Source data are provided with this paper and available via FigShare https://doi.org/10.6084/m9.figshare.26528224. Source data are provided with this paper.

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

## Acknowledgements

We thank Hannah Turner and Will Lessin for electron microscopy support. We thank Charles Bowman and JC Ducom for computational support. We thank Lauren Holden for administrative support. We thank Sai Sundar Rajan Raghavan and Xinglin Yang for valuable discussion. Figure 2B was created with BioRender.com. A.S. acknowledges support from CIRM (EDUC4-12811).

## Author contributions

A.S., A.G., and L.L.L. conceived the project. A.G., A.S., and A.S.T. expressed and purified protein. A.G. prepared cryo-EM samples,

collected and processed cryo-EM data, and built and refined atomic models. A.S. performed biochemical experiments. J.R.L. and M.L.F. performed molecular dynamics simulations. A.G. and A.S. wrote the manuscript with input and edits from L.L.L. and A.B.W.

## Competing interests

L.L.L. is an inventor on patents that describe the discovery of synthetic STING agonists. The four patents are as follows: "Agonists of Stimulator of Interferon Genes STING" Petrassi, M. Lairson, L.L., Chin, E., Schultz, P.G., Yu, C., Yang, B., Heather, V., Grant, S., Li, Y, Pacheco, A., Chu, A. Johnson, K. Chatterjee, A. U.S. Patent Application 62/706,683. September 2, 2020."Bicyclic agonists of stimulator of interferon genes sting" Petrassi, M. Yu, C., Wang, J., Chatterjee, Schultz, P.G., Johnson, K., Chu, A., Chin, E., Lairson, L.L. U.S. Patent Application 62/889,679. August 21, 2019."Monocyclic agonists of stimulator of interferon genes sting" Petrassi, M., Yu, C., Wang, J., Chatterjee, A. Yu, C., Chatterjee, A. Gamo Albero, A.M., Gupta, A., Tamiya, J., Schultz, P.G., Johnson, K., Chu, A., Chin, E., Lairson, L.L. U.S. Patent Application 62/889,669. August 9, 2020."6(1H-imidazolyl)pyridazine compounds as agonists of stimulator of interferon genes sting" Lairson, L.L., Chin, E., Chatterjee, A., Kumar, M., Gamo Albero, A., Petrassi, M., Schultz, P., Yu, C., Tamiya, J., Vernier, W., Gupta, A. Modukuri, R. US Patent Application 62/633,409. February 21, 2018.The remaining authors declare no competing interests.
