## [Transparent Peer Review file · Nature Communications]

Distinct oligomeric assemblies of STING induced by non-nucleotide agonists

Corresponding Author: Dr Andrew Ward

Version 0:

Reviewer comments:

Reviewer #1

(Remarks to the Author)

This manuscript contributed by Gharpure et al reports multiple cryo-EM structures of oligomeric STING in complex with non-nucleotide agonists, SR-717 or diABZI, in the presence of C53. They found that the STING/SR-717/C53 complex forms curved monolayer oligomers. Surprisingly, the STING/diABZI/C53 complex mainly forms bilayer oligomers without obvious curvature, along with a part of curved oligomers. The STING dimer units in these STING/diABZI/C53 structures are activated forms with LBD in open conformation. To handle the heterogeneity of the non-curved bilayer oligomers, they presented two different structures, namely the “together” and “apart” conformations. They also investigated the curved STING/diABZI/C53 complex using 3DVA and proposed that curvature would be a trigger for disruption of the bilayer assembly to expose a monolayer of STING oligomers. The bilayer assembly induced by the agonistic diABZI are interesting and unusual because the bilayer structure of apo chicken STING is proposed as an autoinhibited state (PDB: 8IK0; Molecular Cell, 2023). They proposed that the bilayer assembly of STING bound to diAZBI mediated by LBD-LBD interactions has functional importance for regulating STING activation rates. By performing functional assays, they concluded that diAZBI has slower activation rates compared to SR-717. The authors proposed that the mixture of bilayer and monolayer assembly in these diABZI structures are a semi-autoinhibited state with delayed activation kinetics.

Overall, this manuscript raised an important topic regarding the complexity of STING activation mode induced by different types of agonists and provided a structural basis for the distinctions. Some important points remain unclear in the current manuscript. Some paragraphs and figures are hardly to understand and should be improved.

Major points:

1. The strategy for including C53 in the cryo-EM analysis of STING/SR-717 and STING/diABZI is unclear. To my knowledge, C53 facilitates the efficient formation of the oligomers. This point is important as the authors proposed that the head-to-head interactions visualized in the STING/diABZI/C53 complex have functional relevance to induce a slower activation rate. Therefore, the authors need to test whether oligomers mediated by head-to-head interaction can also be seen with STING/diABZI complex in the absence of C53, because the functional assay were performed in the absence of C53.

2. Lack of sufficient illustration for the structures:

a, Page 13: ‘The head-to-head interface is mediated by a conserved LP motif (L225 and P226 in human STING) that docks into a hydrophobic pocket comprised of the LBD α 1 helix and the base of the lid region on the opposing layer.’ It is hardly to understand based solely on the presentation of the manuscript. The appropriate figure should be included. The author may consider providing more labels and showing the LP motif and α 1 helix in Fig.4B or elsewhere.

b, Page 13: ‘Comparison of the “together” and “apart” conformations reveal that the head-to-head interfaces between different copies of STING are quite heterogeneous and symmetric’.

Although movie S1 may convey this point, a figure comparing the heterogeneous head-to-head interfaces is necessary.

c, Fig.4G: I am confused whether this figure is based on the structure or MD simulation or both. I suppose that most side chains here are structural models rather than the determined structure? As the authors mentioned the resolving of N187 and N188 once, but the latter R220, F221, D223, E246, Q252 and R253 seem to be the prediction model. If so, the authors need to note this point in the figure and its legend.

d, Page 16: ‘This “open” conformation is also similar to HB3089 complex (44.5 Å) but is contracted when compared to the crystal structure of the isolated LBD with diABZI compound 2 (52.7 Å).’ The appropriate figure should be included.

3. As the authors stated ‘Within the LBD there is asymmetric density corresponding to the diABZI ligand with a conspicuous

tail for the morpholinopropoxyl group (Fig. 5A, Fig. S4A). This is an interesting observation considering that STING is a homodimeric protein that should be able to bind asymmetric molecules in two equivalent orientations.' Regarding the asymmetric binding mode of diABZI inside the homo-dimeric LBD, do all the sites show similar asymmetric densities of diABZI? The binding sites seem to be quite heterogeneous as the author stated 'Interestingly, density for R238 is only observed in a subset of STING-diABZI complexes between the curved, together, and apart structures.' This could be a result of averaging two different conformations. According to Table S1, the 'curved', 'together' and 'apart' conformations contain two, four and six copies of diABZI, respectively. Do all these densities show features of the morpholinopropoxyl group for confident modeling? The authors should explain in more details how they handle the potential mixture of two orientations of diABZI during model building the oligomeric diABZI structures. Besides, have the authors tried applying symmetry when refining these structures?

4. From the current data, it is still not exclusive whether the difference in the activation rates of SR-717 and diABZI is simply due to differences in the close and open conformations of LBDs, or to differences in their higher oligomerization states of monolayer and bilayer.

Minor points:

1. diABZI-3 is more appropriate than diABZI for the compound name.

2. Fig.S4 and Fig.5: The authors need to specify which diABZI structure and its cryo-EM map are used for presentation for these figures, at least in the legend. Related to this point, the authors seem not fully mention the visualization of diABZI and C53 in each structure, although this information could be indirectly picked from Table S1.

3. The figures are quite difficult to understand due to the lack of necessary labels, and should be extensively revised. I have listed some but not limited to these.

Fig.S2D-E: please add labels for $\alpha 2$, $\alpha 3$ and $\alpha 2$ - $\alpha 3$ loop

Fig.S2F: please add labels for each TMs, at least in one protomer

Fig.S4E: please add labels for Q273, S275 and others

Fig.4A-D: please inform the location of the LP motif

Fig.4E-G: please add labels for the $\beta 2$, $\beta 3$ strands and post- $\beta 3$ loop.

Fig.5A: please add lines for hydrogen bonds

Fig.S4A, 5A: please add labels for the morpholinopropoxyl group

Fig.4C and Fig.5A,D: Better to add labels for the LBD lid region. When disordered, dashed line helps.

Fig.S4C-D: please add labels for $\alpha 1$ helices and the distances between them

Fig.S5E-F: please provide labels for the residues being mentioned in the main text

Fig.S5E: please provide labels for hydrophilic section and aromatic section, respectively. This figure is hard to understand. As hydrophilic and aromatic residues are colored blue and yellow, respectively, what are residues in cyan and green colors stands for? The surface representation here however brings little information but confusion, unless improved for presentation.

4. Referenced structures should be provided with their PDB IDs: Fig.1D, 6B, 6D

5. References 41 and 44 are the same paper, merge them.

Reviewer #2

(Remarks to the Author)

Overall Assessment:

This work is highly significant to immunology and related fields such as cancer immunotherapy and drug development. It offers novel insights into the structural mechanisms underlying STING (Stimulator of Interferon Genes) activation by non-nucleotide agonists, demonstrating how distinct agonists induce unique conformations and oligomeric states of STING.

The paper specifically investigates STING activation by non-nucleotide agonists, including SR-717 and diABZI, using cryo-electron microscopy (cryo-EM). The structural data reveal key differences in oligomeric assembly, which may have important therapeutic implications. However, some discussion and clarification of the methodology are necessary to fully substantiate the claims and maximize the impact of the findings. With these revisions, the study holds strong potential for publication.

Significant Findings:

1. Structural Insights into STING Activation:

- The study shows that STING's conformation and oligomerization are strongly influenced by the specific ligand binding. SR-717 induces a closed conformation, leading to lateral packing and curved oligomers, similar to the natural agonist 2',3'-cGAMP. On the other hand, diABZI induces an open conformation, resulting in linear, double-stranded oligomers with limited curvature.

- These conformational shifts have significant implications for downstream signaling. The closed conformation induced by SR-717 promotes faster STING activation, while diABZI's open conformation slows phosphorylation, which may be advantageous for immune response modulation.

2. Kinetic and Functional Implications:

- The distinct kinetic profiles of STING activation by SR-717 and diABZI present opportunities for selective agonist development. SR-717's rapid activation may be ideal for scenarios requiring strong immune stimulation, while diABZI's slower kinetics could be better suited for conditions where immune modulation is preferable.
- The head-to-head interactions observed in diABZI-bound STING are particularly noteworthy, as they suggest partial autoinhibition. However, this hypothesis remains speculative and would benefit from additional functional validation.

Suggested Revisions:

1. Clarification of Cryo-EM Data Interpretation:

- The paper relies heavily on cryo-EM for structural insights, but the heterogeneous nature of the assemblies observed (e.g., "together" and "apart" conformations) might introduce ambiguity in interpretation. The conclusions about distinct oligomeric states and their functional implications seem well-supported, but further validation, such as biochemical or in vivo functional assays, could strengthen the claims.

2. Molecular Dynamics (MD) and Membrane Simulation:

- Modeling Missing Residues: The paper does not detail how missing residues in the cryo-EM data were handled in the MD simulations. Providing more information on how these residues were modeled would increase the reliability of the simulations.
- Double Membrane Layers: A clear explanation of how the double membrane layers in the head-to-head oligomers were generated is lacking. Clarifying the methodology, including the use of tools like CHARMM-GUI, would improve the understanding of the simulation approach.

3. Quantitative Water Pore Analysis:

- The role of C53 in regulating water molecules within the proton pore is mentioned, but the figures (especially Figure S5) do not clearly show differences between the presence and absence of C53. A more quantitative analysis, such as counting the number of water molecules in the pore, would strengthen the argument and provide better support for the conclusions.

4. Generalization of Findings:

- The study focuses solely on SR-717 and diABZI, representing two distinct structural outcomes. However, it is essential to consider whether these results can be generalized to other STING agonists. Expanding the analysis to include other agonists or discussing the broader applicability of the findings would enhance the study's impact.

Conclusion and Impact:

The paper offers novel structural insights into the mechanisms of STING activation, emphasizing how ligand-induced conformational changes and oligomer assembly affect activation kinetics. While these structural findings are significant, incorporating the suggested revisions will further enhance the paper's impact, particularly in advancing therapeutic applications for immune modulation.

In summary, the data analysis and conclusions are generally robust, though certain speculative elements could benefit from additional experiments or further discussion to reinforce the study's overall impact.

Reviewer #3

(Remarks to the Author)

In this manuscript "Distinct oligomeric assemblies of STING induced by non-nucleotide agonists", Gharpure and colleagues show the cryo-EM structures of STING with two STING agonists, SR-717 and diABZI, which bind to LBD of STING and induce close and open conformational changes of LBD lid, respectively. Both SR-717 and diABZI induced 180° rotation of the LBD. Notably, they show that the two agonists displayed different cryo-EM patterns: the structure of STING+SR-717+C53 is similar with STING+cGAMP+C53, whereas the structure of STING+ diABZI is more dynamics and complicated. The authors demonstrate that the open-LBD state induced by diABZI leads to partially autoinhibited state and a delayed activation kinetics of STING compared with SR-717. The findings that different non-nucleotide agonists induce distinct oligomeric assemblies of STING are interesting.

Major concerns:

1. The authors show that diABZI induced a range of conformational states and different degrees of curvature (Fig 3 C-E). However, the relevance of the different curvature with STING activation needs to be defined.
2. The authors compared SR-717 (10 μ M) vs diABZI (50 nM) using THP1 cells (Fig 3G and 3H). The delayed p-STING induced by diABZI is not so significant. The half an hour delay could be caused by the low dose of diABZI, or the different rate of cell transmission of compounds. Could the authors compare different concentrations of compounds in different cell lines? The IFN-beta protein expression could also be measured in these experiments. The current data is not strong enough to support the conclusion that diABZI triggers a delayed activation kinetics of STING compared with SR-717.

How about the activation kinetics of STING in cGAMP treatment group compared with diABZI and SR-717?

3. Could the authors analyze the activation of STING-TBK1-IRF3 signaling and the STING oligomeric assemblies using native gel assay?

4. The R238 data in Fig 5 is interesting. Could the authors mutate R238 to see the STING-TBK1-IRF3 phosphorylation cascades and innate immune response to characterize the role of this residue in diABZI-mediated STING signaling?
5. Since SR-717 and diABZI induce different STING oligomeric assemblies, I wonder whether they would induce different innate immune response, for example, would genome wide analysis of the transcriptional response be different? Recent studies show that STING agonist induce the expression of interleukin (IL)-35 and interleukin (IL)-18 to negatively or positively regulate anti-tumor activity. Whether SR-717 and diABZI would induce distinct gene expression?
6. Previous cryo-EM study of STING-HB3089 suggests that the diABZI analog does not induce 180°-rotation of the LBD. How do the authors speculate the different results? Would the C53 binding in the TMD induce the LBD 180°-rotation?
7. The current discussion needs to be strengthened.

Version 1:

Reviewer comments:

Reviewer #1

(Remarks to the Author)

The revised manuscript is improved and the authors have addressed all of the concerns/suggestions.

Reviewer #2

(Remarks to the Author)

The revised manuscript makes a significant and novel contribution to the structural understanding of STING activation by non-nucleotide agonists. Through a comprehensive analysis combining cryo-EM and molecular dynamics simulations, the authors provide compelling insights into the distinct oligomeric assemblies induced by SR-717 and diABZI-3. Their findings elucidate critical differences in ligand-induced STING conformations and their functional implications, offering a deeper mechanistic understanding of STING activation.

The authors have thoroughly addressed all previous reviewer concerns, substantially improving the manuscript's clarity, methodological rigor, and interpretability. The addition of further structural analyses, biochemical validation experiments, and methodological refinements has significantly enhanced the study's robustness. Notably, the revised discussion presents a more cohesive synthesis of how ligand-specific oligomerization states influence STING activation kinetics, highlighting key implications for therapeutic strategies targeting the STING pathway.

Overall, this study represents a substantial advancement in STING structural biology and innate immune signaling. The findings provide valuable insights for the rational design of STING-targeted therapeutics. Given the high novelty, methodological rigor, and translational significance of this work, I strongly support its publication in its current form.

Reviewer #3

(Remarks to the Author)

The revised manuscript has addressed all my concerns.

Response to reviewers:

We thank all reviewers for their careful reading of the manuscript and for their valuable comments and questions on this study. We believe that by addressing these comments we have strengthened our manuscript. Attached below are point-by-point responses to all comments.

Reviewer #1 (Remarks to the Author):

This manuscript contributed by Gharpure et al reports multiple cryo-EM structures of oligomeric STING in complex with non-nucleotide agonists, SR-717 or diABZI, in the presence of C53. They found that the STING/SR-717/C53 complex forms curved monolayer oligomers. Surprisingly, the STING/diABZI/C53 complex mainly forms bilayer oligomers without obvious curvature, along with a part of curved oligomers. The STING dimer units in these STING/diABZI/C53 structures are activated forms with LBD in open conformation. To handle the heterogeneity of the non-curved bilayer oligomers, they presented two different structures, namely the “together” and “apart” conformations. They also investigated the curved STING/diABZI/C53 complex using 3DVA and proposed that curvature would be a trigger for disruption of the bilayer assembly to expose a monolayer of STING oligomers. The bilayer assembly induced by the agonistic diABZI are interesting and unusual because the bilayer structure of apo chicken STING is proposed as an autoinhibited state (PDB: 8IK0; Molecular Cell, 2023). They proposed that the bilayer assembly of STING bound to diAZBI mediated by LBD-LBD interactions has functional importance for regulating STING activation rates. By performing functional assays, they concluded that diAZBI has slower activation rates compared to SR-717. The authors proposed that the mixture of bilayer and monolayer assembly in these diABZI structures are a semi-autoinhibited state with delayed activation kinetics.

Overall, this manuscript raised an important topic regarding the complexity of STING activation mode induced by different types of agonists and provided a structural basis for the distinctions. Some important points remain unclear in the current manuscript. Some paragraphs and figures are hardly to understand and should be improved.

Major points:

1. The strategy for including C53 in the cryo-EM analysis of STING/SR-717 and STING/diABZI is unclear. To my knowledge, C53 facilitates the efficient formation of the oligomers. This point is important as the authors proposed that the head-to-head interactions visualized in the STING/diABZI/C53 complex have functional relevance to induce a slower activation rate. Therefore, the authors need to test whether oligomers mediated by head-to-head interaction can also be seen with STING/diABZI complex in the absence of C53, because the functional assay were performed in the absence of C53.

We thank the reviewer for this comment. Since its identification, C53 has proven to be an effective tool to help induce oligomerization for structural studies of human STING. The inclusion of C53 was used to determine the initial structures of the human STING-

cGAMP oligomer (Lu et al., 2022; PMID: 35388221). In this study, the authors showed that using C53 and cGAMP together induced robust oligomerization in an in vitro assay, while either ligand alone did not. It is important to note that in a later study, Liu et al., 2023 were able to resolve an oligomeric structure of human STING bound to cGAMP without C53 (PMID: 37086726). Importantly, the structures of human STING/cGAMP and human STING/cGAMP/C53 show no structural differences as we note in line 112. Furthermore, the addition of C53 promoted oligomerization and substantially improved map quality of human STING in complex with cGAMP and a molecular glue-like compound (Li et al., 2024; PMID: 37828400). We thus included C53 in our sample preparation for this study to improve oligomerization and map quality.

To address this reviewer comment, we collected an additional cryo-EM dataset of human STING with diABZI-3 without C53, prepared using the same protocol as the structures in the manuscript. The majority of particles in this dataset failed to oligomerize (representative micrograph showing predominantly STING dimers below on the left) and we were unable to resolve a high-order oligomeric structure. However, we did sparingly see evidence of STING bilayers, consistent with the diABZI-3/C53 dataset (circled in red).

To address the important concern raised about the comparison of C53-based structural data with functional assay data collected in the absence of C53, we have completed additional studies and now include phospho-STING Western blot and ISRE-Luciferase reporter data collected in the presence of C53 in the manuscript (Fig. S5, see below). C53 accelerates the rate of downstream pathway activation in both assay formats. Importantly, consistent with our initial observations made in the absence of C53, in the presence of C53 we observe delayed activation kinetics with diABZI-3 compared to SR-717.

Figure S5. Rates of SR-717- and diABZI-3-dependent STING activation in the presence of C53

2. Lack of sufficient illustration for the structures:

a, Page 13: ‘The head-to-head interface is mediated by a conserved LP motif (L225 and P226 in human STING) that docks into a hydrophobic pocket comprised of the LBD α 1 helix and the base of the lid region on the opposing layer.’. It is hardly to understand based solely on the presentation of the manuscript. The appropriate figure should be included. The author may consider providing more labels and showing the LP motif and α 1 helix in Fig.4B or elsewhere.

We have added a supplementary figure (Fig. S6) to clarify the position of the LP motif in the autoinhibited structure and our diABZI-3 structures.

b, Page 13: ‘Comparison of the “together” and “apart” conformations reveal that the head-to-head interfaces between different copies of STING are quite heterogeneous and symmetric’.

Although movie S1 may convey this point, a figure comparing the heterogeneous head-to-head interfaces is necessary.

We have added an additional figure panel (Fig. S6E) containing the superposition of the LBDs from these two conformations and also refer to the local resolution figures in panels S3C-D to highlight the lower resolution at the head-to-head interfaces.

c, Fig.4G: I am confused whether this figure is based on the structure or MD simulation or both. I suppose that most side chains here are structural models rather than the determined structure? As the authors mentioned the resolving of N187 and N188 once, but the latter R220, F221, D223, E246, Q252 and R253 seem to be the prediction model. If so, the authors need to note this point in the figure and its legend.

We apologize for the lack of clarity. We used MD to identify residues that are potentially involved in mediating the head-to-head interactions. We then show these residues on the determined structure in Fig. 4G. The figure legend has been updated to clarify this point.

d, Page 16: 'This "open" conformation is also similar to HB3089 complex (44.5 Å) but is contracted when compared to the crystal structure of the isolated LBD with diABZI compound 2 (52.7 Å).' The appropriate figure should be included.

We have added additional figure panels comparing the "open" conformations in Fig. S4E-F.

3. As the authors stated 'Within the LBD there is asymmetric density corresponding to the diABZI ligand with a conspicuous tail for the morpholinopropoxyl group (Fig. 5A, Fig. S4A). This is an interesting observation considering that STING is a homodimeric protein that should be able to bind asymmetric molecules in two equivalent orientations.' Regarding the asymmetric binding mode of diABZI inside the homo-dimeric LBD, do all the sites show similar asymmetric densities of diABZI? The binding sites seem to be quite heterogeneous as the author stated 'Interestingly, density for R238 is only observed in a subset of STING-diABZI complexes between the curved, together, and apart structures.' This could be a result of averaging two different conformations. According to Table S1, the 'curved', 'together' and 'apart' conformations contain two, four and six copies of diABZI, respectively. Do all these densities show features of the morpholinopropoxyl group for confident modeling? The authors should explain in more details how they handle the potential mixture of two orientations of diABZI during model building the oligomeric diABZI structures. Besides, have the authors tried applying symmetry when refining these structures?

We thank the reviewer for highlighting this point. All diABZI-3 structures were refined without symmetry imposed due to slight asymmetries in the protein conformation and to try and accurately model the morpholinopropoxyl groups in the orthosteric ligand. In all copies of diABZI-3, we see clear density for a morpholinopropoxyl tail attached to the dimeric amidobenzimidazole moiety. This tail density is unambiguously stronger and connected to the ligand on one side allowing for confident modeling in nearly all twelve copies of diABZI-3 with the sole exception being one of the peripheral copies in the "open" conformation which has lower local resolution in the LBD. We have now included an additional supplementary figure (Fig. S7) to show all diABZI-3 densities. Nevertheless, as the reviewer mentions, there could be heterogeneity with a mixture of both orientations, and it remains unclear what the functional implications for

directionality of the ligand could be. We have updated the text at line 304 to take this potential heterogeneity into account:

“Within the LBD we observed clear density for the dimeric amidobenzimidazole moiety corresponding to the core of diABZI-3 ligand along with asymmetric density for the morpholinopropoxyl tail. This suggests a preferential binding orientation of the asymmetric ligand in the LBD, which is an interesting observation considering that STING is a homodimeric protein that should be able to bind asymmetric molecules in two equivalent orientations. However, the functional implications of ligand directionality remain unclear, and it is possible that there is a mixture of both orientations in each LBD.”

4. From the current data, it is still not exclusive whether the difference in the activation rates of SR-717 and diABZI is simply due to differences in the close and open conformations of LBDs, or to differences in their higher oligomerization states of monolayer and bilayer.

We thank the reviewer for highlighting this uncertainty and we agree that both possibilities (or a combination thereof) could contribute to the observed differences in rates. Indeed, we believe that the observed differences in activation kinetics between the ligands is due to both differences in the LBD conformations and the formation of higher oligomeric states. We hypothesize that the range of oligomeric states observed with diABZI-3 may be derived from a less effective ability of open LBD ligands to fully disrupt the auto-inhibited apo-oligomer, which leads to the observed unique oligomeric state of STING bound to diABZI-3. In the discussion section, we have revised the text to make this distinction clearer as follows (line 418):

“We believe that the range of oligomeric states observed with diABZI-3 may be derived from a less effective ability of open-LBD ligands to fully disrupt the auto-inhibited apo-oligomer. Furthermore, this head-to-head conformational arrangement of STING induced by diABZI-3 likely impedes the formation of curved oligomers, which could delay or modulate the tone or strength of downstream pathway activation. Consistent with this potential model, based directly on TBK1-dependent STING phosphorylation status or induced levels of IRF reporter signal, delayed rates of STING activation were observed for diABZI-3 when compared to SR-717 in functional assays. Importantly, observed kinetic differences could also directly result from the structural differences of the closed and open conformations of LBDs and indeed our observations could arise from a combination of both effects.”

Minor points:

1. diABZI-3 is more appropriate than diABZI for the compound name.

We have replaced diABZI with diABZI-3 when specifically discussing diABZI compound 3 and use diABZI to discuss the general linked amidobenzimidazole class of compounds.

2. Fig.S4 and Fig.5: The authors need to specify which diABZI structure and its cryo-EM map are used for presentation for these figures, at least in the legend. Related to this

point, the authors seem not fully mention the visualization of diABZI and C53 in each structure, although this information could be indirectly picked from Table S1.

We have now added which conformations are used for figure panels in the legends. As the reviewer notes, we were able to resolve and model one copy of diABZI-3 and one copy of C53 in each STING dimer that was fully built (2 in the curved conformation, 4 in the together conformation, and 6 in the apart conformation).

3. The figures are quite difficult to understand due to the lack of necessary labels, and should be extensively revised. I have listed some but not limited to these.

Fig.S2D-E: please add labels for $\alpha 2$, $\alpha 3$ and $\alpha 2$ - $\alpha 3$ loop

Done- we also added labels for the $\alpha 1$ helix and distances in Fig. S2D as well as residue labels in Fig. S2E.

Fig.S2F: please add labels for each TMs, at least in one protomer

Done- labels added.

Fig.S4E: please add labels for Q273, S275 and others

Done- residue labels added.

Fig.4A-D: please inform the location of the LP motif

We have added Fig. S6A-D to illustrate the location of the LP motif in autoinhibited and diABZI-3 bound structures.

Fig.4E-G: please add labels for the $\beta 2$, $\beta 3$ strands and post- $\beta 3$ loop.

We have added labels in Fig. 4G for clarity.

Fig.5A: please add lines for hydrogen bonds

Done.

Fig.S4A, 5A: please add labels for the morpholinopropoxyl group

Done.

Fig.4C and Fig.5A,D: Better to add labels for the LBD lid region. When disordered, dashed line helps.

Done.

Fig.S4C-D: please add labels for $\alpha 1$ helices and the distances between them

Done.

Fig.S5E-F: please provide labels for the residues being mentioned in the main text

Done.

Fig.S5E: please provide labels for hydrophilic section and aromatic section, respectively. This figure is hard to understand. As hydrophilic and aromatic residues are colored blue and yellow, respectively, what are residues in cyan and green colors stands for? The surface representation here however brings little information but confusion, unless improved for presentation.

We have updated the figure for clarity and now show hydrophilic residues mentioned in the text in the figure (now Fig. S8E) and aromatics and histidines at the base of the TMD in Fig. S8F.

4. Referenced structures should be provided with their PDB IDs: Fig.1D, 6B, 6D

PDB IDs have been added to all figure legends.

5. References 41 and 44 are the same paper, merge them.

Thank you for bringing this error to our attention, which has been corrected in the revised version of the manuscript.

Reviewer #2 (Remarks to the Author):

Overall Assessment:

This work is highly significant to immunology and related fields such as cancer immunotherapy and drug development. It offers novel insights into the structural mechanisms underlying STING (Stimulator of Interferon Genes) activation by non-nucleotide agonists, demonstrating how distinct agonists induce unique conformations and oligomeric states of STING.

The paper specifically investigates STING activation by non-nucleotide agonists, including SR-717 and diABZI, using cryo-electron microscopy (cryo-EM). The structural data reveal key differences in oligomeric assembly, which may have important therapeutic implications. However, some discussion and clarification of the methodology are necessary to fully substantiate the claims and maximize the impact of the findings. With these revisions, the study holds strong potential for publication.

Significant Findings:

1. Structural Insights into STING Activation:

- The study shows that STING's conformation and oligomerization are strongly influenced by the specific ligand binding. SR-717 induces a closed conformation, leading to lateral packing and curved oligomers, similar to the natural agonist 2',3'-cGAMP. On the other hand, diABZI induces an open conformation, resulting in linear, double-stranded oligomers with limited curvature.

- These conformational shifts have significant implications for downstream signaling. The closed conformation induced by SR-717 promotes faster STING activation, while diABZI's open conformation slows phosphorylation, which may be advantageous for immune response modulation.

2. Kinetic and Functional Implications:

- The distinct kinetic profiles of STING activation by SR-717 and diABZI present opportunities for selective agonist development. SR-717's rapid activation may be ideal for scenarios requiring strong immune stimulation, while diABZI's slower kinetics could be better suited for conditions where immune modulation is preferable.

- The head-to-head interactions observed in diABZI-bound STING are particularly noteworthy, as they suggest partial autoinhibition. However, this hypothesis remains speculative and would benefit from additional functional validation.

Suggested Revisions:

1. Clarification of Cryo-EM Data Interpretation:

- The paper relies heavily on cryo-EM for structural insights, but the heterogeneous nature of the assemblies observed (e.g., "together" and "apart" conformations) might introduce ambiguity in interpretation. The conclusions about distinct oligomeric states and their functional implications seem well-supported, but further validation, such as biochemical or in vivo functional assays, could strengthen the claims.

We thank the reviewer for this comment. We have added additional biochemical validation in Fig. S5 and in responses to reviewer 3. We agree that in vivo assays would add to the findings of our study, but these are beyond the scope of our current manuscript. We believe that our structural findings and biochemical validation provide important insight into the complexity of STING activation modes and will serve as a foundation for further studies from our group and others.

2. Molecular Dynamics (MD) and Membrane Simulation:

- Modeling Missing Residues: The paper does not detail how missing residues in the cryo-EM data were handled in the MD simulations. Providing more information on how these residues were modeled would increase the reliability of the simulations.

We thank the reviewer for this comment. We added a sentence in line 534 to clarify: "We modeled the missing residues in the cryo-EM structure using the modeling/building missing loops tool in MOE (CCG) and prepared the structures for the simulations using MOE."

- Double Membrane Layers: A clear explanation of how the double membrane layers in the head-to-head oligomers were generated is lacking. Clarifying the methodology, including the use of tools like CHARMM-GUI, would improve the understanding of the simulation approach.

For the head-to-head oligomers, we only used one lipid bilayer, as we truncated the second molecule to the head to calculate the head-to-head interactions, which is now clarified in line 536.

3. Quantitative Water Pore Analysis:

- The role of C53 in regulating water molecules within the proton pore is mentioned, but the figures (especially Figure S5) do not clearly show differences between the presence and absence of C53. A more quantitative analysis, such as counting the number of water molecules in the pore, would strengthen the argument and provide better support for the conclusions.

We thank the reviewer for this comment. To quantify the difference, we calculated the water distribution density along the z-axis of the pore using a Kernel-density estimation and use these plots in the figure (now Fig. S8).

4. Generalization of Findings:

- The study focuses solely on SR-717 and diABZI, representing two distinct structural outcomes. However, it is essential to consider whether these results can be generalized

to other STING agonists. Expanding the analysis to include other agonists or discussing the broader applicability of the findings would enhance the study's impact.

We thank the reviewer for this insight and agree that the evaluation of additional STING agonists would enhance our understanding of these findings. As described in response to point 2 of reviewer 3 below, comparisons using an alternative open conformation-inducing ligand (i.e., cyclic diGMP) is challenged by the observed lack of cell-based potency in our assay system, which likely results from the introduced confounding need for active uptake for the natural CDN ligands. This manuscript serves as an initial insight into the structural heterogeneity of STING oligomers in response to different stimuli through the use of representative small molecule agonists of both open and closed conformation-inducing ligand classes. We have clarified in the discussion that further generalization of these findings will require the structural investigation of a wider variety of agonists (i.e cyclic dinucleotides, synthetic agonists and TMD binders).

Line 456: "In this light, further expansion of these analyses to a wide variety of STING agonists (i.e., synthetic agonists, CDNs, and non-LBD binders) will be required, in order to fully understand the functional relevance of the unique oligomeric state we observe here in the context of a synthetic open conformation-inducing LBD ligand."

Conclusion and Impact:

The paper offers novel structural insights into the mechanisms of STING activation, emphasizing how ligand-induced conformational changes and oligomer assembly affect activation kinetics. While these structural findings are significant, incorporating the suggested revisions will further enhance the paper's impact, particularly in advancing therapeutic applications for immune modulation.

In summary, the data analysis and conclusions are generally robust, though certain speculative elements could benefit from additional experiments or further discussion to reinforce the study's overall impact.

Reviewer #3 (Remarks to the Author):

In this manuscript "Distinct oligomeric assemblies of STING induced by non-nucleotide agonists", Gharpure and colleagues show the cryo-EM structures of STING with two STING agonists, SR-717 and diABZI, which bind to LBD of STING and induce close and open conformational changes of LBD lid, respectively. Both SR-717 and diABZI induced 180° rotation of the LBD, Notably, they show that the two agonists displayed different cryo-EM patterns: the structure of STING+SR-717+C53 is similar with STING+cGAMP+C53, whereas the structure of STING+ diABZI is more dynamics and complicated. The authors demonstrate that the open-LBD state induced by diABZI leads to partially autoinhibited state and a delayed activation kinetics of STING compared with

SR-717. The findings that different non-nucleotide agonists induce distinct oligomeric assemblies of STING are interesting.

Major concerns:

1. The authors show that diABZI induced a range of conformational states and different degrees of curvature (Fig 3 C-E). However, the relevance of the different curvature with STING activation needs to be defined.

Previous studies had suggested that the positive curvature induced by activated STING may contribute to anterograde transport to the Golgi (Lu et al., 2022; PMID: 35388221 and Liu et al., 2023; PMID: 37086726). By resolving different populations with varying degrees of curvature, we speculate that diABZI-3-bound STING may be subject to additional regulation. In this scenario, the STING molecules with pronounced curvature in Fig. 3C would support COP II recruitment and vesicle budding similar to SR-717 and cGAMP-bound STING, whereas flattened STING molecules in the bilayer (Fig. 3E) may be inhibited from doing so. We have added more detail in the text at line 228 to better explain this hypothesis:

“These results suggest that while diABZI-3 can induce similar curvature to closed-LBD agonists such as SR-717 (Fig. 3C), the constraints from the bilayer may partially restrict the ability of diABZI-3-bound STING to form curved oligomers. The flattened diABZI-3-bound STING molecules in the bilayer (Fig. 3E) may prevent COPII recruitment and anterograde transport which could play an additional regulatory role in a mechanism that serves to modulate rates of STING activation and/or the strength of downstream pathway signaling by this class of ligands.”

2. The authors compared SR-717 (10 μ M) vs diABZI (50 nM) using THP1 cells (Fig 3G and 3H). The delayed p-STING induced by diABZI is not so significant. The half an hour delay could be caused by the low dose of diABZI, or the different rate of cell transmission of compounds. Could the authors compare different concentrations of compounds in different cell lines? The IFN-beta protein expression could also be measured in these experiments. The current data is not strong enough to support the conclusion that diABZI triggers a delayed activation kinetics of STING compared with SR-717.

How about the activation kinetics of STING in cGAMP treatment group compared with diABZI and SR-717?

We thank the reviewer for this comment and understand potential concern surrounding the differences in ligand concentrations that were used to make comparisons (10 μ M SR-717 versus 50 nM diABZI-3). These concentrations were chosen based on observed potency and efficacy levels in both the luciferase reporter and phospho-STING Western blot assays. Specifically, we compared relative kinetics at concentrations that elicit matched maximal levels of STING phosphorylation. Lower concentrations of SR-717 result in sub-optimal levels phospho-STING activation that is

difficult to detect in a way that can be compared to optimal diABZI-3 levels. Further, inconsistent and variable levels of pathway activation are observed using super-physiological concentrations of diABZI-3, which we believe likely results from the limited solubility of this dimeric agonist. For these reasons, we believe that the chosen concentrations (~EC80 concentrations that induce the same degree of phospho-STING activation) provides the best and most reliable conditions for comparing relative activation rates. To address your valid concern, for review purposes only, below we include representative data acquired at higher concentrations of diABZI-3 (1 μ M). Importantly, at this concentration when matched maximal levels of STING phosphorylation are observed the timing remains consistent with 50 nM diABZI-3. However, for the reasons stated above, we believe the published observations should be those observed using the physiologic and fully soluble concentration originally used.

We completely agree that IFN β is a valid orthogonal readout for cGAS-STING activation. However, we observe that STING phosphorylation status and luciferase reporter data provide the most consistent readouts for comparing relative kinetics, given that IFN β activation is downstream, transient (compared to reporter signal) and more variable. To address your point, we include IFN β activation data in THP-1s (shown below but for review only).

It is challenging to compare the relative kinetics of downstream pathway activation induced by an alternative open conformation-inducing agonist, as these are cyclic

dinucleotides (CDNs) (e.g., cyclic diGMP) that rely on active transport. The transporter for cGAMP is known to be SLC46A2 (Cordova et al., 2021; PMID: 34235268), however that of cyclic diGMP is still unknown and an inability to assess differences in relative abundance of these transporters and associated rates of import confound the interpretation of observed results. For instance, our studies with cyclic diGMP suggest that it has an EC₅₀ around 100 μ M in the ISRE-luciferase assay (data shown below), whereas the reported value in digitonin-permeabilized cells is around 500 nM (Zhang et al., 2014; PMID: 23747010). For these reasons, we are unable to compare the relative kinetics of pathway activation between CDN and synthetic ligands.

To address your concern surrounding the cell permeability of the synthetic agonists used in our studies, in the revised version of the manuscript, we now include phospho-STING western data collected in the presence of C53 (Supplemental Figure 4, shown below). The observed ability of C53 to induce observable STING phosphorylation within 30 minutes for both ligands (with the relative rates observed for LBD ligands being maintained), strongly suggests that rates of SR-717 and diABZI-3 transmission into the cell do not contribute to the observed differences in activation kinetics in the absence of C53.

Supplemental Figure S5. Rates of SR-717- and diABZI-3-dependent STING activation in the presence of C53

Nevertheless, your collective points are very well taken. Given the technical challenges and limitations associated with these assays, we agree that claims surrounding relative activation kinetics and their correlation to the primary finding of this manuscript (i.e., the observed newly identified head-to-head oligomeric form of STING uniquely observed in the presence of diABZI compared to SR-717 obtained using saturating ligand concentrations) need to be tempered with limitations of interpretation clearly stated. To address this, we have modified the appropriate sections of text in the revised version of the manuscript to temper these claims and explicitly state the rationale for the concentrations of ligands and the importance of expanding these analyses to other STING agonists in the future.

Line 235: “We evaluated ligand class-dependent differences in relative rates of STING activation, by assessing STING (S366) phosphorylation status in THP-1 cells (Fig. 3G, H). When evaluated at \sim EC₈₀ concentrations that induce the same degree of pathway activation, diABZI-3 was found to activate STING at slower rate, with peak levels of STING phosphorylation occurring \sim 1 hour later for diABZI-3 when compared to SR-717 (2-4 hours versus 1-2 hours of stimulation, Fig. 3G, H). Notably, given the disparity in cell-based potencies and limitations of solubility at super-physiological concentrations of the dimeric diABZI-3 ligand, the evaluated concentrations in this assay are physiologically most relevant. Nonetheless, the potential caveat of these concentration differences should be noted.”

3. Could the authors analyze the activation of STING-TBK1-IRF3 signaling and the STING oligomeric assemblies using native gel assay?

We thank the reviewer for this comment and have investigated STING oligomeric states using a native gel assay. Indeed, Liu *et al.* (2023) employed the use of native gels in their identification of the autoinhibited state of apo-STING (PMID: 37086726). In their reported studies they observed the presence of STING oligomers in the absence of cGAMP (i.e., the apo form of STING), which led to their closer investigation of apo-STING via cryo-EM. Similarly, in our analysis using native gels we observed the presence of STING oligomers in the absence of any ligand. Additionally, we observed that both SR-717 and diABZI (at 100 and 30 μ M, respectively) induce the formation of oligomers (see below), as expected. Our cryo-EM studies enabled the further, higher resolution, investigation of the differences in these oligomeric states.

4. The R238 data in Fig 5 is interesting. Could the authors mutate R238 to see the STING-TBK1-IRF3 phosphorylation cascades and innate immune response to characterize the role of this residue in diABZI-mediated STING signaling?

We thank the reviewer for this suggestion. While we believe that, given the well-established role of this residue in the literature, at least one copy engaging with the ligand is essential for pathway activation by ligands that bind the LBD of STING (Gao *et al.*, 2013; PMID: 23910378, Shang *et al.*, 2019, PMID: 30842659), we generated a suite of mutants in order to investigate both the head-to-head interaction of diABZI-bound STING and R238. Unfortunately, despite extensive effort, we were unable to generate a stable STING knock-in THP-1 cell line containing an R238A mutation at this time. This was in contrast to our ability to perform a small alanine scan of residues H185, Y186, N187, and N188, which are all involved in the diABZI-STING head-to-head interface. Our investigation of diABZI-dependent phospho-STING activation over time did not yield observable differences with these mutants (representative data shown below). We anticipate that delineating the role of specific residues involved in open-LBD binding, as

well as those that mediate the distinct head-to-head interface in diABZI-bound STING, will be the focus of a subsequent manuscript, and in the interest of time we seek to have our primary significant new findings published as soon as possible.

5. Since SR-717 and diABZI induce different STING oligomeric assemblies, I wonder whether they would induce different innate immune response, for example, would genome wide analysis of the transcriptional response be different? Recent studies show that STING agonist induce the expression of interleukin (IL)-35 and interleukin (IL)-18 to negatively or positively regulate anti-tumor activity. Whether SR-717 and diABZI would induce distinct gene expression?

These proposed experiments represent an excellent direction, which we intend to pursue as a next step toward developing our understanding of the role of these differing STING oligomers in downstream signaling. However, significant follow up studies would be required to investigate genome-wide transcriptional responses to both ligands at multiple timepoints, which we anticipate and hope to be the primary content for a subsequent manuscript.

6. Previous cryo-EM study of STING-HB3089 suggests that the diABZI analog does not induce 180°-rotation of the LBD. How do the authors speculate the different results? Would the C53 binding in the TMD induce the LBD 180°-rotation?

We believe that the differences between our diABZI-3 structures and the previously reported HB3089 structures are due to the oligomeric states of the structures, specifically the lack of lateral oligomerization of STING molecules in the HB3089 structure. As shown in Fig. 6D, lateral oligomerization of STING dimers requires a conformational change in the LBD $\alpha 2$ - $\alpha 3$ loop to avoid a steric clash. This movement would result in a potential clash with the connector helix in the “crossover” conformation seen in apo and HB3089-bound STING. Thus, we believe that rearrangement of the connector helix and consequently the 180° rotation of the LBD is a result of higher-order oligomerization, which are both considered hallmarks of activated STING. We have added more detail to explain this in the paragraph beginning at line 354. If the authors of the HB3089 study had been able to induce higher-order oligomerization, potentially through the use of other detergents such as DDM/CHS or with the inclusion of C53, we speculate that they would also observe the 180° rotation of the LBD.

7. The current discussion needs to be strengthened.

We thank the reviewer for this very appropriate comment. In the revised version of the manuscript, we have significantly expanded our discussion section.

Response to reviewers:

We thank all reviewers for their careful reading of the manuscript and for their valuable comments and questions on this study.

Reviewer #1 (Remarks to the Author):The revised manuscript is improved and the authors have addressed all of the concerns/suggestions.

Reviewer #2 (Remarks to the Author):The revised manuscript makes a significant and novel contribution to the structural understanding of STING activation by non-nucleotide agonists. Through a comprehensive analysis combining cryo-EM and molecular dynamics simulations, the authors provide compelling insights into the distinct oligomeric assemblies induced by SR-717 and diABZI-3. Their findings elucidate critical differences in ligand-induced STING conformations and their functional implications, offering a deeper mechanistic understanding of STING activation. The authors have thoroughly addressed all previous reviewer concerns, substantially improving the manuscript's clarity, methodological rigor, and interpretability. The addition of further structural analyses, biochemical validation experiments, and methodological refinements has significantly enhanced the study's robustness. Notably, the revised discussion presents a more cohesive synthesis of how ligand-specific oligomerization states influence STING activation kinetics, highlighting key implications for therapeutic strategies targeting the STING pathway. Overall, this study represents a substantial advancement in STING structural biology and innate immune signaling. The findings provide valuable insights for the rational design of STING-targeted therapeutics. Given the high novelty, methodological rigor, and translational significance of this work, I strongly support its publication in its current form.

Reviewer #3 (Remarks to the Author):The revised manuscript has addressed all my concerns.